# Improving rechargeable magnesium batteries through dual cation co-intercalation strategy

Ananyo Roy[1], Mohsen Sotoudeh[2], Sirshendu Dinda [1], Yushu Tang[3,4], Christian Kübel [1,3,4], Axel Groß [1,2], Zhirong Zhao-Karger [1,3], Maximilian Fichtner[1,3] & Zhenyou Li [1,5,6,7] ✉

The development of competitive rechargeable Mg batteries is hindered by the poor mobility of divalent Mg ions in cathode host materials. In this work, we explore the dual cation co-intercalation strategy to mitigate the sluggishness of $Mg^{2+}$ in model $TiS_2$ material. The strategy involves pairing $Mg^{2+}$ with $Li^+$ or $Na^+$ in dual-salt electrolytes in order to exploit the faster mobility of the latter with the aim to reach better electrochemical performance. A combination of experiments and theoretical calculations details the charge storage and redox mechanism of co-intercalating cationic charge carriers. Comparative evaluation reveals that the redox activity of $Mg^{2+}$ can be improved significantly with the help of the dual cation co-intercalation strategy, although the ionic radius of the accompanying monovalent ion plays a critical role on the viability of the strategy. More specifically, a significantly higher $Mg^{2+}$ quantity intercalates with $Li^+$ than with $Na^+$ in $TiS_2$. The reason being the absence of phase transition in the former case, which enables improved $Mg^{2+}$ storage. Our results highlight dual cation co-intercalation strategy as an alternative approach to improve the electrochemical performance of rechargeable Mg batteries by opening the pathway to a rich playground of advanced cathode materials for multivalent battery applications.

Low-cost and sustainable energy storage systems are required to keep up with the increasing energy demands of today's society[1-3]. In that context, battery chemistries based on metallic negative electrode (anode) and multivalent ion shuttle meets the criterion of high energy density and has generated substantial interest amongst researchers[4,5]. Li ion batteries (LIBs) are the benchmark for practical performance metrics[6-9] but multivalent systems have shown promise as viable candidates that could supplement the energy demands in the future by employing comparatively safe metal anodes[10-12]. Rechargeable magnesium batteries (RMBs), where Mg metal is used as the negative electrode due to its high volumetric capacity (3833 mAh $L^{-1}$) and low tendency to form dendrites, have attracted particular attention[13-15]. The low redox potential of Mg (−2.37 V vs SHE) and divalent charge carriers offer high theoretical energy densities[15]. However, research has shown that the knowledge gained from LIBs cannot be directly translated in a facile manner to RMB

[1]Helmholtz Institute Ulm (HIU), Helmholtzstraße 11, 89081 Ulm, Germany. [2]Institute of Theoretical Chemistry, Universität Ulm, Oberberghof 7, 89081 Ulm, Germany. [3]Institute of Nanotechnology (INT), Karlsruhe Institute of Technology (KIT), Hermann-von-Helmholtz-Platz 1, 76344 Eggenstein-Leopoldshafen, Germany. [4]Karlsruhe Nano Micro Facility (KNMF), Karlsruhe Institute of Technology (KIT), Eggenstein-Leopoldshafen, Germany. [5]Qingdao Institute of Bioenergy and Bioprocess Technology, Chinese Academy of Sciences, No. 189 Songling Road, Laoshan District, Qingdao, Shandong 266101, China. [6]Shan-dong Energy Institute, Qingdao 266101, China. [7]Qingdao New Energy Shandong Laboratory, Qingdao 266101, China. ✉e-mail: zhenyou.li@kit.edu

systems due to various properties associated with multivalent chemistries[15,16].

One major challenge is the lack of appropriate intercalation materials for positive electrode or cathode[17]. Contrary to the monovalent Li ions, the solid-state diffusion of Mg$^{2+}$ is sluggish due to the strong Coulombic interaction with the host material. In effect, solid-state diffusion is typically the rate-limiting step for redox reactions, thus rendering the conventional intercalation approach as a hurdle for RMBs. Different strategies have been employed to counter this bottleneck, the most popular being particle downsizing[16,17]. This approach is an effective way to shorten the diffusion path length, unlike the longer diffusion path in bigger particles which can restrict the intercalating ion to the particle surface and leave the core unreacted, as explained by the core-shell particle model[18].

Vacancy-mediated diffusion mechanism showed improved kinetics in non-stoichiometric anatase TiO$_2$, where Ti vacancies were synthetically generated by fluorine doping which in turn created additional charge storage sites[19]. This strategy highlighted how defect engineering can be influential in promoting Mg ion intercalation. In a different work Li et al. demonstrated the strategy of intercalating solvated Mg$^{2+}$[20]. In this approach, 1,2 dimethoxyethane (DME) coordinated Mg$^{2+}$ [Mg$^{2+}$•3DME] enabled fast solid-state diffusion by lowering the charge density of the Mg ions. The strategy of cation-solvent co-intercalation greatly enhanced the reaction kinetics and indicated that the intercalation of two species in tandem can be a viable approach to promote fast kinetics in the positive electrode[20].

In a similar direction, Li et al. demonstrated dual-cation co-intercalation in the Chevrel phase Mo$_6$S$_8$ cathode by employing a Mg-Li dual salt electrolyte[21,22]. The working principle was based on the 'rocking chair-type' model, where both charge carriers (Li$^+$ and Mg$^{2+}$) participated in the half-cell reactions, as illustrated in Fig. 1. It was observed that Mo$_6$S$_8$ could accommodate both carrier ions in equal concentrations during discharge and seemingly co-deposited Mg and Li in a dendrite free manner during charge[21]. Co-intercalation of both carrier ions delivered a higher insertion voltage compared to the single carrier Mg analogue which could indicate improved insertion kinetics. First-principles calculations explained that the solid-state diffusion of both carrier ions was concomitant in nature, where the ions moved along the reaction path via a coordinated interaction[22]. According to the authors this type of coordinated motion greatly reduced the Mg$^{2+}$ migration barrier (-0.2 eV) compared to the individual hopping mechanism, which showed a much higher migration barrier (-0.55 eV) for Mg ions[22]. Consequently, by virtue of coordinated or concerted interaction between Mg and Li ions, the energy barrier along the diffusion path was reduced considerably, which facilitated faster solid-state diffusion in Mo$_6$S$_8$.

However, after the successful demonstration in Mo$_6$S$_8$, the cation co-intercalation strategy was rarely reported in other host structures[23]. As Chevrel phase Mo$_6$S$_8$ is almost the only material that exhibits fast Mg$^{2+}$ mobility at room temperature (25 °C), the feasibility to transfer the cation co-intercalation concept to other materials remains unexplored. More importantly, it is still not clear whether co-intercalation with a monovalent ion could indeed improve the overall Mg ion storage capacity. Additionally, the storage mechanism, redox chemistry and other key factors governing Mg storage need to be clarified for better understanding and is necessary for further advancements.

In this work we attempted to answer these questions by establishing a model system with a layered host (TiS$_2$) as the positive electrode, an advanced dual salt electrolyte based on fluorinated alkoxyborate and Mg metal as the negative electrode. Based on that, a multimodal approach combining experimental and theoretical techniques was applied to understand how the size of different monovalent charge carriers (Li$^+$/Na$^+$) along with the thermodynamics and structural stability of the host compound affect the viability of cation co-intercalation. In order to assess the influence of monovalent ions on the reactivity of Mg ions, a holistic combination of elemental, structural and redox probes was used together with standard electrochemical techniques. The elemental analysis highlighted the direct impact of the size of the monovalent ion on the intercalation of the divalent Mg ions. Structural characterization of cycled TiS$_2$ along with density functional theory (DFT) studies demonstrated the importance of thermodynamic and structural stability of the host compound to reversibly accommodate more than one cationic charge carrier. Especially for layered materials which are susceptible to phase change due to the sliding of layers, phase transformation plays an active role in the co-intercalation of different carrier ions. Furthermore, spectroscopic probing through ex situ electron energy-loss spectroscopy (EELS) of cycled TiS$_2$ positive electrodes helped to clarify the redox mechanism.

## Results
### Establishing model systems for co-intercalation and the impact of monovalent ions
To demonstrate the versatility of cation co-intercalation, a typical layered TiS$_2$ was selected as the host material for establishing model systems. In order to investigate the size effect of monovalent ions on co-intercalation, Mg electrolyte was mixed in stoichiometric amounts with the corresponding Li and Na counterparts, separately. For the convenience of the reader, the cell system with the dual salt electrolyte 0.3 M Mg[B(hfip)$_4$]$_2$ – 0.15 M Li[B(hfip)$_4$] / DME, ([B(hfip)$_4$]$^-$ = hexafluoroisopropyloxy borate ion) has been referred to as the Mg-Li system and the cell system with 0.3 M Mg[B(hfip)$_4$]$_2$ – 0.15 M Na[B(hfip)$_4$] / DME as the Mg-Na system. In this section we have

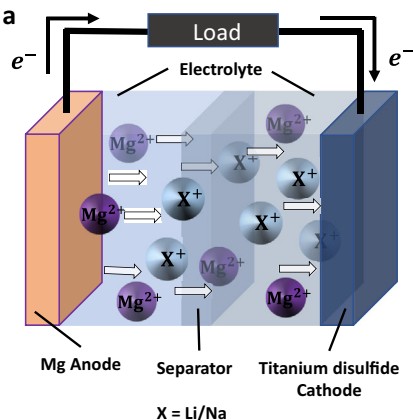
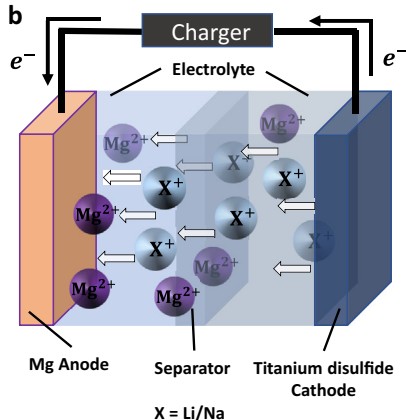

**Fig. 1 | Schematic of the working principle.** Co-intercalation of divalent Mg and monovalent Li/Na-ions for the dual salt electrolyte system with Mg metal anode and TiS$_2$ cathode during (**a**) discharge and (**b**) charge.

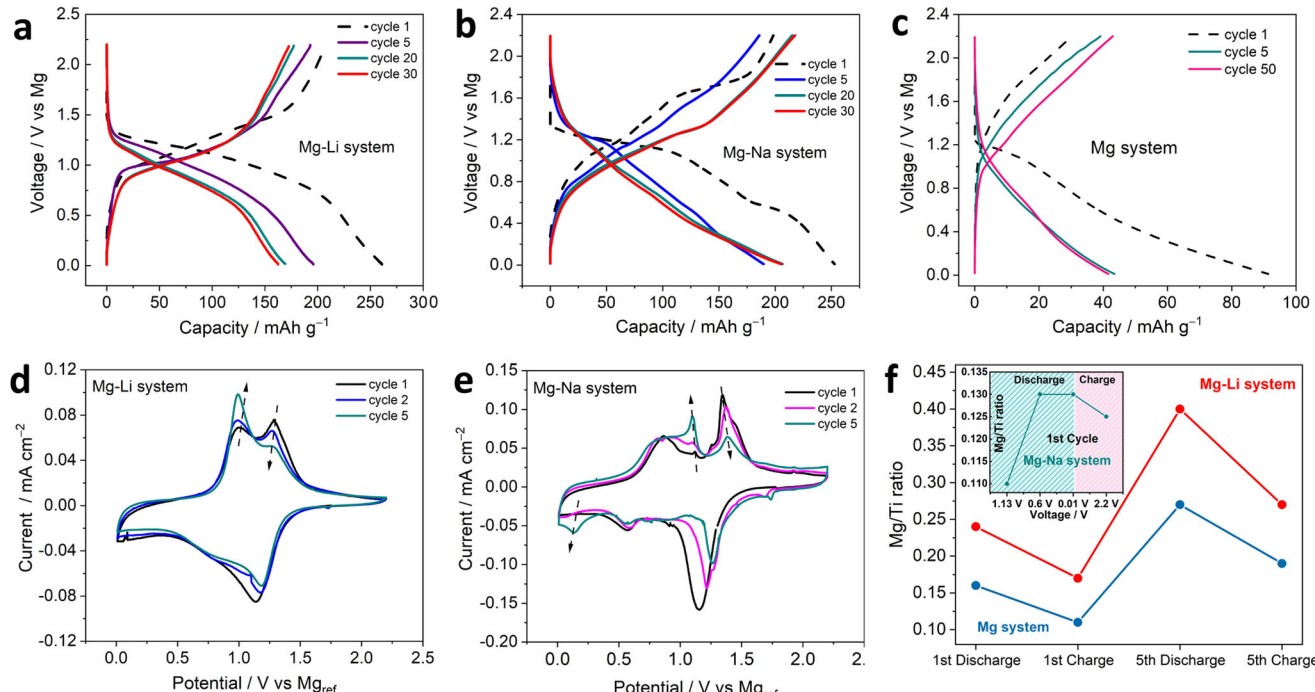

**Fig. 2 | Comparing dual cation systems against the reference Mg system using electrochemical and elemental techniques as follows.** Two-electrode galvanostatic charge-discharge profiles of TiS$_2$ in (**a**) Mg-Li system, (**b**) Mg-Na system and (**c**) Mg system cycled between 0.01 V and 2.2 V vs Mg with a constant current of 20 mA g$^{-1}$. Three-electrode cyclic voltammograms (CV) of TiS$_2$ for (**d**) Mg-Li system and (**e**) Mg-Na system operated between 0.01 V and 2.2 V vs Mg reference (Mg$_{ref}$) scanned with 0.1 mV s$^{-1}$. **f** Mg/Ti ratio in Mg-Li (red curve), Mg (blue curve) and Mg-Na (inset) cells calculated from ICP-OES data.

analyzed the two dual salt systems (Mg-Li and Mg-Na) looking into the co-intercalation of cationic charge carriers. In the process we individually validated both systems by combining electrochemical and elemental probes. The outcome of the investigations clarified the impact of two different monovalent ions on the degree of co-intercalation.

Mg-Li system: The Mg-Li cell delivered a reasonably high initial discharge capacity of ~250 mAh g$^{-1}$ (Fig. 2a), which agreed closely with the theoretical capacity of TiS$_2$ (239 mAh g$^{-1}$) for 1e$^-$ exchange. In contrast, for the single salt Mg cell, TiS$_2$ cathode delivered a much lower initial discharge capacity of ca. 90 mAh g$^{-1}$ (Fig. 2c). The poor performance of the single salt Mg cell did not match up with the findings reported by the Sun et al. [24] where an initial discharge capacity of ~ 250 mAh g$^{-1}$ was delivered. The reason behind this could be that the measurements were conducted at an elevated temperature of 60 °C[24]. Furthermore, the single salt Li system delivered ~200 mAh g$^{-1}$ (Supplementary Fig. 1a) in the voltage range (1.5 V – 2.9 V vs Li) corresponding to the intercalation regime[25]. The Mg-Li system produced a slightly higher initial discharge capacity compared to the Li system. The corresponding charge capacities of the Mg-Li and the single ion Mg systems were 206 mAh g$^{-1}$ and 40 mAh g$^{-1}$, respectively. The irreversible capacity loss observed after the first cycle in the Mg-Li cell (ca. 30 mAh g$^{-1}$) was slightly reduced, compared to the Mg cell (ca. 40 mAh g$^{-1}$). The irreversible capacity loss is a direct consequence of carrier ion entrapment in the TiS$_2$ crystal structure[24,26]. The Mg-Li system delivered a reversible capacity of ~140 mAh g$^{-1}$ after 100 cycles (Supplementary Fig. 2b) as opposed to 40 mAh g$^{-1}$ delivered by the Mg system (Supplementary Fig. 2a). In addition to improved specific capacity, the Mg-Li cell had a higher insertion voltage of ~1.25 V and a higher nominal voltage of ~1.1 V with a smooth plateau-like profile during the first discharge, similar to the Li system (Supplementary Fig. 1a). The Mg cell on the contrary showed a steep drop in the voltage profile (Fig. 2c) with a low nominal voltage of ~0.6 V. Moreover, the Mg-Li system exhibited reasonable rate performance by delivering a discharge capacity of ~157 mAh g$^{-1}$ at C/4, ~144 mAh g$^{-1}$ at C/2, ~131 mAh g$^{-1}$ at 1 C and

~115 mAh g$^{-1}$ at 2 C (Supplementary Fig. 2c). The above observations highlighted that the insertion kinetics was significantly improved in the dual salt Mg-Li system in comparison to the single salt Mg system, whereby the sluggishness of Mg$^{2+}$ affected the kinetics adversely.

The redox activity of the Mg-Li system was further characterized by three-electrode cyclic voltammetry (CV), as shown in Fig. 2d, and the voltammogram showed a prominent reduction peak centered around 1.14 V vs Mg$_{ref}$, which corroborated reasonably well with the nominal voltage. The prominence of the reduction peak in the Mg-Li system suggested that the intercalation kinetics improved significantly compared to the Mg system, which had a broad and asymmetric CV profile (Supplementary Fig. 3a) and in addition had low current density, characteristic of sluggish mobility of Mg$^{2+}$. The corresponding anodic scan involved a two-step oxidation process (Fig. 2d) centered around ~1.0 V and ~1.4 V, which indicated that two types of charge carriers occupied the storage sites with different potentials through structural relaxation[22,27]. Furthermore, the voltammogram of the Li system (Supplementary Fig. 3b) had a completely different redox profile with the redox couple positioned around 2.33 V vs Li$_{ref}$ (1.66 V vs Mg). The fundamental differences in the profile shape and redox peak position between the two voltammograms could be explained by the co-intercalation of Li$^+$ and Mg$^{2+}$ in TiS$_2$ in the dual salt Mg-Li system and intercalation of only Li$^+$ in the single salt Li system. The evolution of the oxidation peaks with cycling (Fig. 2d) showed a trend, whereby the intensity increased for the peak at ~1.0 V, while the peak at ~1.4 V lost intensity. The electrochemical activity of the storage site corresponding to 1.4 V was observed to decrease during cycling. However, the precise nature of this phenomenon remains unclear and warrants further investigation in future studies.

The improved electrochemical performance of TiS$_2$ in the Mg-Li system raised a question about the capacity share of each cationic charge carrier. Li ions being more mobile can either facilitate or hinder co-intercalation of sluggish Mg ions[21,22,28]. Hence, it was of importance that the individual charge compensations of the respective charge

carriers ($Mg^{2+}$ and $Li^+$) were evaluated in order to obtain a clear picture of the actual redox activity of $Mg^{2+}$. This was done by estimating the concentration of the intercalated carrier ions with the help of inductive coupled plasma optical emission spectroscopy (ICP-OES). After the first discharge, a Mg/Ti atomic ratio of ~0.24 was observed along with a Li/Ti value of ~0.35 (Fig. 2f and Supplementary Table 1). In comparison, the Mg/Ti ratio after first discharge in the single salt Mg system was ~0.16 (Fig. 2f). The improvement in Mg intercalation was accompanied by a comparable amount of Li in the Mg-Li system. In terms of charge transfer, $Mg^{2+}$ and $Li^+$ contributed 0.48 $e^-$ and 0.35 $e^-$, respectively, which underlined their equitable nature to charge distribution and capacity share. However, the first cycle showed significant Mg trapping, with an atomic fraction of ~0.17 Mg per formula unit of $TiS_2$ being irreversibly trapped (Fig. 2f). Qualitatively, scanning transmission electron microscopy electron dispersive X-ray (STEM-EDX) mapping also confirmed co-intercalation of Mg ions in the Mg-Li system, as shown by the uniform distribution of Mg in $TiS_2$ after the first discharge (Supplementary Fig. 4d). It should be noted that after the first charge, $TiS_2$ still exhibited significant Mg entrapment, as is apparent from the STEM-EDX map (Supplementary Fig. 4h), which was corroborated by the ICP-OES data. The initial irreversible Mg entrapment in the Mg-Li system is a drawback that was also observed in the Mg system (Fig. 2f) where a fraction of ~0.11 Mg remained in the $TiS_2$ structure. The issue of irreversible Mg trapping has been reported previously[24] and underlines the consequence of poor solid-state diffusion of $Mg^{2+}$. Further investigation after the fifth cycle showed improved $Mg^{2+}$ insertion kinetics as a Mg/Ti atomic ratio of ~0.13 could be reversibly de-/intercalated in the Mg-Li system, whereas an atomic ratio of only ~0.08 was reversibly de-/intercalated in the single salt Mg system (Fig. 2f). Thus, the Mg-Li system showed superior insertion kinetics for divalent Mg ions enabled by dual cation co-intercalation compared to the single salt Mg system. All ICP-OES data of the above-mentioned samples are presented in Supplementary Table 1.

Mg-Na system: In an attempt to examine the versatility of dual-cation co-intercalation, the concept was extended to Mg-Na dual salt system. The main difference between the systems is the larger ionic radius of $Na^+$ (102 pm), which can impact the storage site preference and interaction potential in $TiS_2$. The electrochemical performance showed an initial discharge capacity of ~250 mAh $g^{-1}$ (Fig. 2b), similar to the Mg-Li system. For comparison, the single salt Na system delivered ~237 mAh $g^{-1}$ after the first discharge (Supplementary Fig. 1b). It is worth noting that the initial discharge capacity of the Mg-Na system also matched closely with the work by Bian et al. [29] However, differences between the voltage profiles exist and could be attributed to the chemical modification of $TiS_2$ cathode upon reacting with the borohydride-based dual salt electrolyte, while with the borate-based dual salt electrolyte, $TiS_2$ did not exhibit any chemical change. Following charging, the Mg-Na system provided a charge capacity of ~196 mAh $g^{-1}$, which corresponds to a capacity loss of 54 mAh $g^{-1}$. Irreversible charge carrier entrapment was the likely reason, similar to both Mg-Li and Mg systems. The discharge voltage profile of the Mg-Na system showed two distinct voltage plateaus at ~1.25 V and ~0.6 V, which indicated a two-phase intercalation process. In comparison, the voltage profile of the Mg-Li and Mg systems resembled a solid-solution type behavior without indication of phase transition. Cycling showed a drastic change in the shape of the discharge voltage profile, where the plateaus disappeared completely by the $5^{th}$ cycle. The change in the shape of the voltage profile was likely due to structural degradation, which will be discussed later on.

The CV profile (Fig. 2e) showed a sharp feature of a reduction process around 1.2 V and another low intensity broad peak around 0.6 V during the first cathodic sweep, which agreed well with the corresponding discharge voltage curve. The anodic sweep showed two main oxidation peaks at 0.85 V and 1.35 V, and another small peak at 1.1 V. These profiles hint at multiple phase transitions, which is a typical feature of $Na^+$ intercalation in layered materials. In the following cycles, the oxidation peak at 1.1 V became more prominent, while the peak at 1.35 V lowered in intensity. The cycling induced changes in the oxidation peaks (1.1 V and 1.35 V) exhibited a similar pattern to that observed in the Mg-Li system, thus warranting further perusal. Additionally, a new reduction peak emerged in the $5^{th}$ cycle at 0.14 V which could be attributed to a probable irreversible conversion reaction which could lead to possible structural degradation. The degradation of the structure was also reported elsewhere for Na-ion batteries, where the more complicated phase transition and the presence of unstable intermediate $Na_xTiS_2$ has been mentioned as a possible reason[30].

The elemental analysis of cycled $TiS_2$ from the Mg-Na system via ICP-OES data showed a strong divergence in the results compared to the Mg-Li system (inset of Fig. 2f and Supplementary Table 2). After full discharge the Mg/Ti ratio was ~0.13 while the Na/Ti value was ~0.81. The high Na content is a strong indicator of an un-equitable specific capacity distribution between Mg and Na ions, unlike the Mg-Li system. After the corresponding charge step it was found that almost all of the Mg (Mg/Ti ~0.125) was irreversibly trapped in the $TiS_2$ structure while the $Na^+$ showed much superior reversibility and only had ~0.09 atomic fraction trapped in the structure. The dominant effect of Na ions and the disproportionate nature of charge contribution from the two cations indicated that the size difference between the monovalent and multivalent ions could be a decisive parameter regarding the feasibility and degree of dual-cation co-intercalation in layered $TiS_2$.

## Structural evolution of $TiS_2$ upon cycling

The layered structure of $TiS_2$ gives rise to a discernible structural change as the concentration of intercalating ions increases[24,31]. The layers are held together by weak van der Waals forces which make them susceptible to slide along the 2D plane, changing the stacking order and stimulating phase transformation. Additionally, the layers also expand and contract along the $c$-direction with varying concentration of charge carriers. Co-intercalation of Mg and Li ions showed expansion of the crystal structure, evident from the ex situ XRD pattern shown in Fig. 3a, where the three most intense reflections at (001), (101) and (102) shifted toward lower angles after the first discharge which suggested increased lattice parameter values[24,32]. In this scenario, the (001) reflection at 5.6° in the pristine $TiS_2$ shifted to 5.15° upon discharge, which demonstrated a $d$-space expansion from 5.7 Å to 6.2 Å. In contrast, no clear shift of the reflections was observed when only Mg ions were intercalated into $TiS_2$ (Fig. 3b) in the Mg system. The likely reason for no obvious structural expansion can be attributed to low Mg concentration in the $TiS_2$ crystal structure as a direct consequence of limited $Mg^{2+}$ intercalation in the Mg system. $TiS_2$ showed reasonable structural reversibility in the Mg-Li system, whereby the diffraction pattern of pristine $TiS_2$ was recovered back upon charging. In addition to the structural reversibility, it was also observed that $TiS_2$ did not undergo any phase transformation during de-/intercalation. The crystal structure retained the P3m1 space group (O1 phase) after intercalation and exhibited a solid-solution behavior which agreed well with the slope-like voltage profile (Fig. 2a). Absence of phase transformation ensured no drastic structural alteration and limited stress generation in the positive electrode. The XRD pattern also did not show any reflections that can be attributed to polysulfides and/or Ti metal, indicating that the conversion reaction did not occur and that the reversible capacity was delivered through the intercalation redox mechanism.

In stark contrast, the structural evolution of $TiS_2$ in the Mg-Na system is completely different from the aforementioned Mg-Li system, primarily due to the role played by the bigger Na ion ($r = 104$ pm)[31]. In order to track the evolution that the $TiS_2$ crystal structure underwent with increasing intercalant concentration, ex situ XRD was conducted at different stages of discharge as shown in Fig. 4a, c. The XRD patterns indicated a larger expansion of the crystal structure along the $c$-axis compared to the Mg-Li system. The (001) peak shifted to 4.55° and

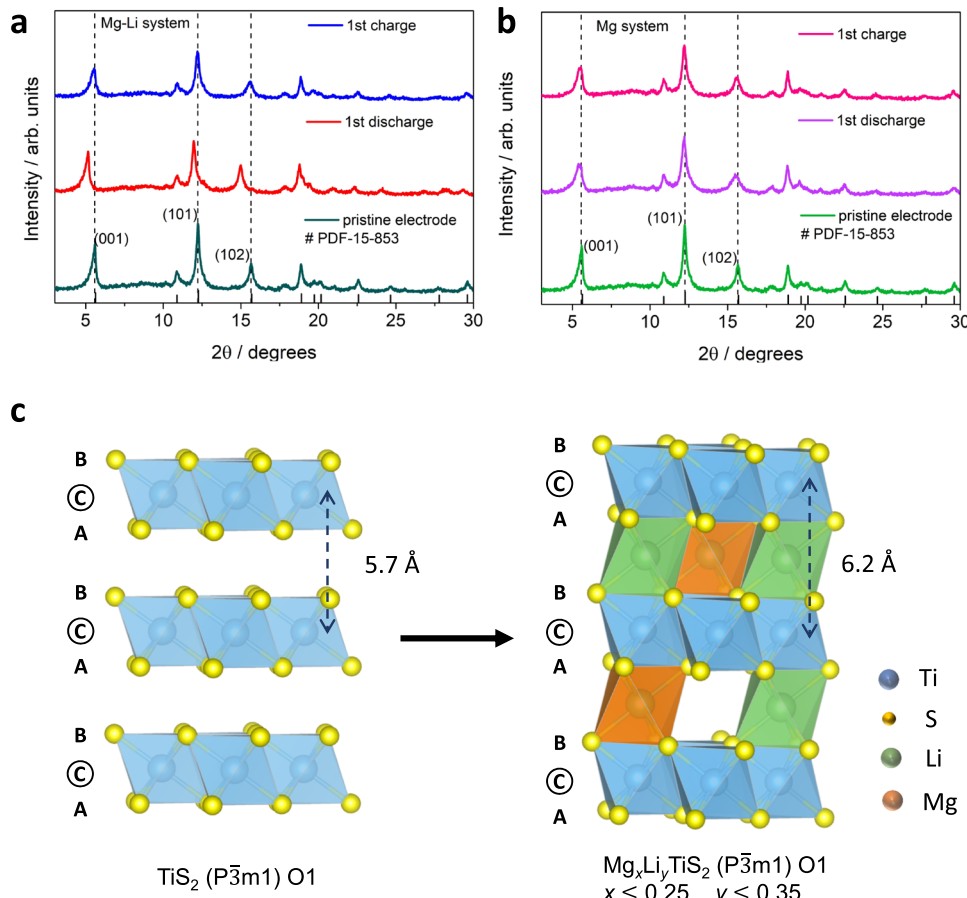

**Fig. 3 | Bulk structure analysis of the cathode in the Mg-Li system upon cycling.** Ex situ XRD of $TiS_2$ comparing the (**a**) Mg-Li system and the reference (**b**) Mg system. **c** Atomic models representing the structural expansion of layered $TiS_2$ in the Mg-Li system upon co-intercalation with no phase transformation.

caused the *d*-space to expand to 7.2 Å upon discharge, significantly larger compared to when the smaller Mg ions ($r = 72$ pm) and Li ions ($r = 76$ pm) were co-intercalated. The interlayer expansion was accompanied by significant change in the crystal structure. The XRD pattern showed broadening upon discharging to 1.13 V accompanied with a low angle shift of the (001) reflection from 5.6° to 4.9°, suggesting the beginning of interlayer expansion. The (101) reflection of the pristine $TiS_2$ appeared broadened while the (102) reflection also dropped in intensity. This could be interpreted as the onset of the formation of a phase different from the pristine O1 phase. It is worth noting that at 1.13 V, $TiS_2$ had a sodium dominant composition of $Mg_{0.11}Na_{0.46}TiS_2$ (inset of Fig. 2f and Supplementary Table 2) which confirmed that the phase transition was driven by Na ion intercalation. Upon further discharge to 0.6 V (also see Fig. 4b), it was observed that the (001) reflection in the pristine O1 phase was replaced by the (003) reflection at 4.55° which corresponded to the prismatic P3 phase (R3m)[31,33].

In conjunction, the appearance of another new reflection at 13.2° indexed as (015) along with the disappearance of the (102) reflection shown in Fig. 4b (also see Fig. 4a), confirmed the formation of the P3 phase. On full discharge (0.01 V), a new reflection (104) emerged at 12.6°, which is characteristic of the O3 phase (R$\bar{3}$m) corresponding to ~1 Na per formula unit of $TiS_2$ ($NaTiS_2$)[31,33]. This agrees closely with the measured stoichiometry ($Mg_{0.13}Na_{0.81}TiS_2$) (Fig. 2f and Supplementary Table 2) as ~0.81 mole fraction of $Na^+$ was intercalated along with a small fraction of $Mg^{2+}$. Hence, a mixture of P3 and O3 phases was formed upon full discharge (see Fig. 4b). However, due to the inadequate resolution quality of the XRD patterns, refinement and phase quantification could not be accomplished.

The low concentration of Mg in the structure could be explained by the stacking order change of layered $TiS_2$, initiated by the intercalation of the Na ions. The phase transformation from O1 to P3 alters the co-ordination geometry of the charge storage site from octahedral to prismatic. The prismatic site prefers the storage of the bigger $Na^+$ as opposed to the smaller $Mg^{2+}$ and $Li^+$[34]. On the contrary, $Mg^{2+}$ finds it favorable to occupy the octahedral site[34]. This site preference mismatch, directly related to phase change, benefitted the storage of the bigger $Na^+$. Thus, making it effectively the dominant charge carrier while suppressing the co-intercalation of $Mg^{2+}$ despite the larger interlayer spacing. It is worth noting that a reasonable reversibility of the structure was observed in the first cycle with the recovery of the pristine O1 (P$\bar{3}$m1) phase upon charging. Almost all of the intercalated Na ions were extracted as per ICP-OES (Supplementary Table 2). However, as discussed in the previous section, the Mg ions were completely trapped and that made the co-intercalation irreversible. Furthermore, the suppression of Mg co-intercalation could also be traced back to the metastable nature of the P3 phase.

The metastability of the P3 phase and its impact on the co-intercalation of Mg will be discussed further with the help of DFT phase diagrams, which will be shown later. The additional investigation of the crystal structure conducted after the second and fifth discharge showed severe amorphization and onset of structural distortions with the eventual disappearance of the characteristic (003) reflection as shown in Supplementary Fig. 5b. This could be another possible reason for the suppressed co-intercalation of Mg ions. The degradation of the structure was corroborated with the drastic change observed in the voltage profile during cycling (Fig. 2b). In stark contrast, $TiS_2$ showed robust structural integrity in the Mg-Li system as the crystallinity and

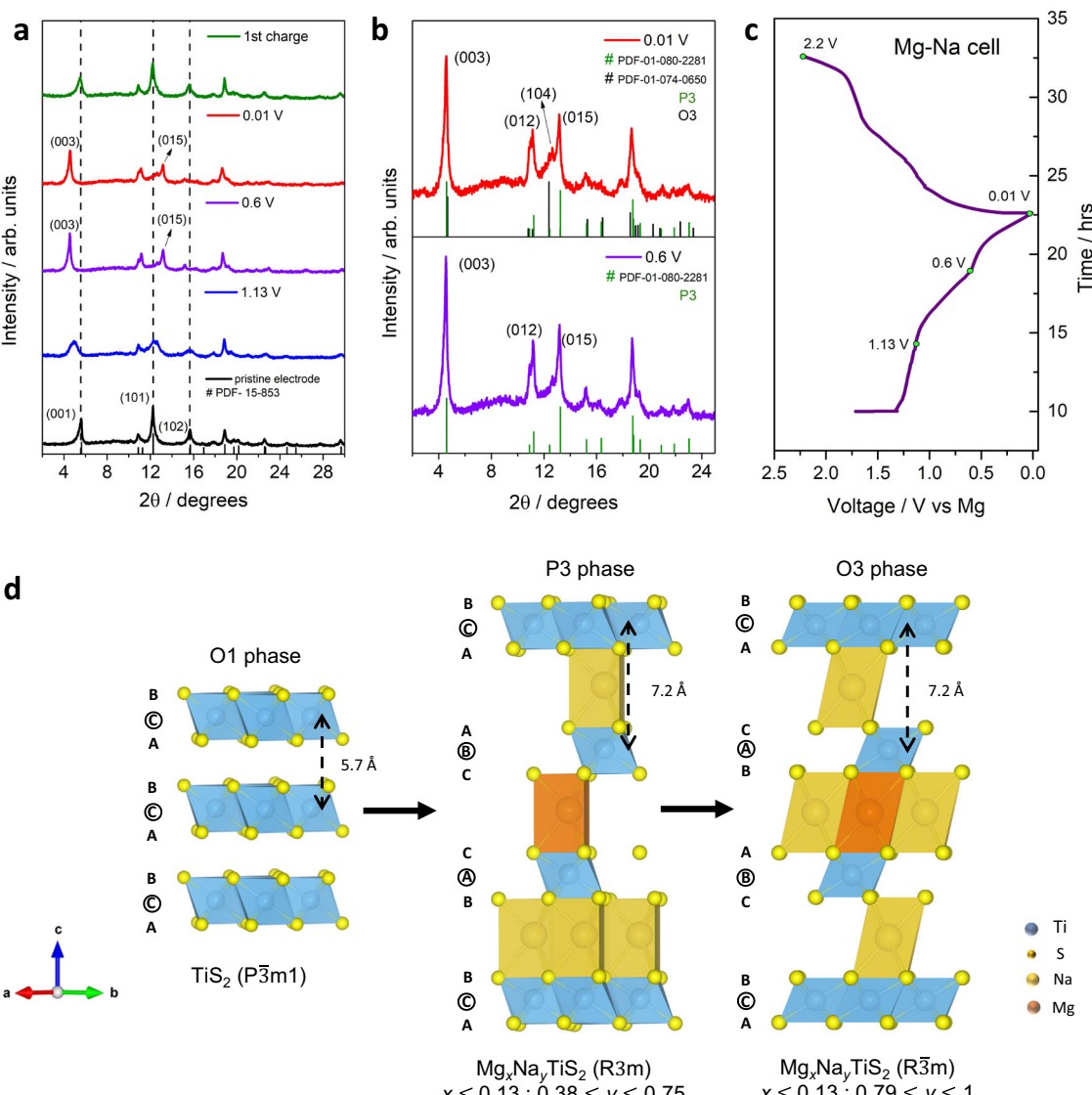

**Fig. 4 | Structural evolution of TiS₂ during co-intercalation of Mg and Na ions.** **a** Ex situ XRD at different states of charge. **b** Magnified XRD pattern at 0.6 V and 0.01 V shows the major P3 phase and the minor O3 phase. **c** Voltage profile showing the different states of charge. **d** Atomic model showing the phase transformation to the Na dominant P3 and O3 mixed phase along with the changes in the stacking sequence of the layers.

O1 phase were retained after multiple cycles as shown in Supplementary Fig. 5a. Based on these findings, a correlation can be established that structural stability could be one of the key parameters required for co-intercalation of two charge carriers.

In conjunction to the bulk structure, it is also important to study the changes the structure undergoes locally upon intercalation of different charge carriers to develop a better understanding of the storage mechanism. Through a combination of ex situ Raman spectroscopy and ex situ transmission electron microscopy (TEM), the local structure of various cycled TiS₂ samples in relation to the aforementioned systems were probed. TiS₂ is a layered material which forms a trigonal structured crystal with a space group of P$\bar{3}$m1 where S-Ti-S slabs interact conventionally through weakly interacting van der Waals forces. On the other hand, single-crystal synchrotron X-ray diffraction measurement suggested the presence of partially covalent Ti-S intralayer interaction and a strong S...S interlayer electron sharing[35], dissimilar to the classical van der Waals force.

The irreducible representation at Γ point is $\Gamma = A_{1g} + E_g + 2A_{2u} + 2E_u$ and the Raman active vibrational modes are the in-plane $E_g$ and out-of-plane $A_{1g}$ symmetric stretching of sulfur atoms (Fig. 5a) The three primary modes of vibrations centered at ~214 cm⁻¹ (assigned to $E_g$), ~331 cm⁻¹ (assigned to $A_{1g}$) and a "shoulder peak" at ~362 cm⁻¹ (herein termed Sh) were measured by Raman spectroscopy of pristine TiS₂ electrode (Fig. 5b)[36]. The origin of the Sh mode of vibration is still under debate as it is linked to several processes such as, presence of excess interlayer titanium[37], van der Waals forces between the interlayers[38] or stress induced forbidden $A_{2u}$ mode. The presence of $E_g$, $A_{1g}$ and Sh suggested that the bulk TiS₂ was trigonal and contained multiple layers in agreement with existing reports[38,39]. Monitoring the collective changes in the vibrational bands can shed light on the induced local structural and environmental changes in TiS₂ following intercalation of charge carriers.

Raman spectra were acquired from the surfaces of pristine TiS₂, Mg-TiS₂ (first discharge), Mg-Li-TiS₂ (first discharge) and Mg-Na-TiS₂ (first discharge) electrodes. No obvious phase change was observed in TiS₂ when Mg ions were intercalated in the single salt Mg system, as the Raman band features resembled closely to those of pristine TiS₂. On the contrary, co-intercalation of Mg and Li ions broadened the main

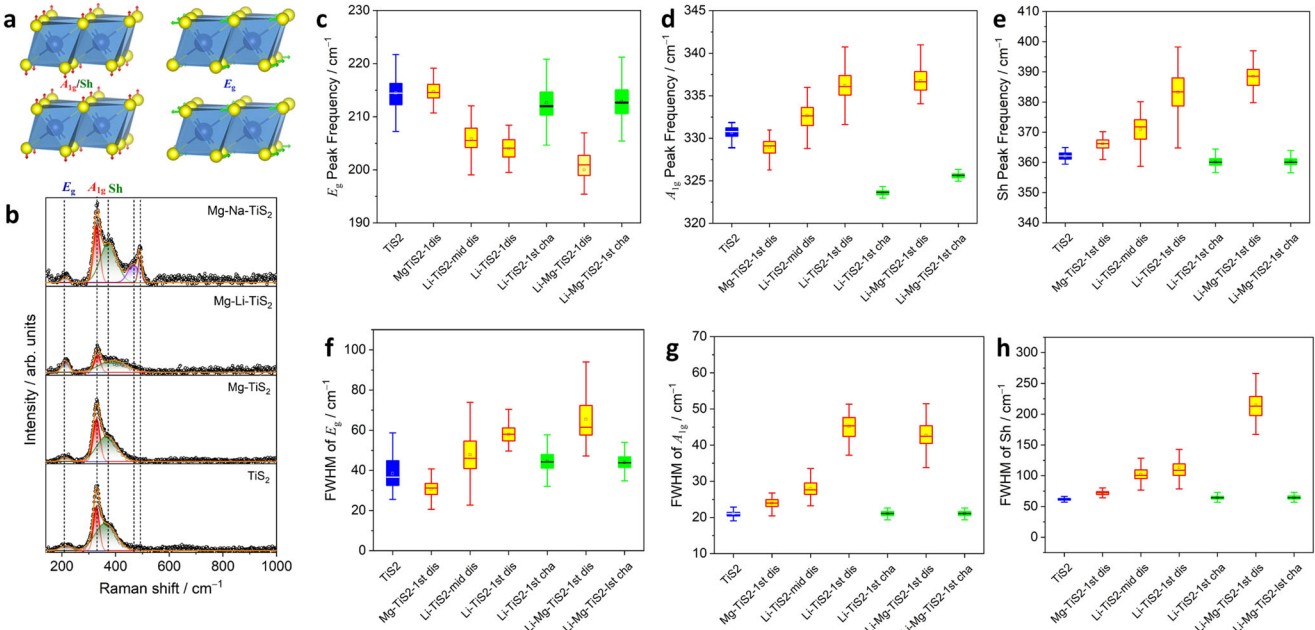

**Fig. 5 | Ex situ Raman spectroscopy of TiS₂. a** Schematic of Raman active $A_{1g}$, Sh and $E_g$ modes of vibration. **b** Raman spectra showing the $A_{1g}$, Sh and $E_g$ vibrational frequencies of TiS₂ in pristine and discharged (Mg-TiS₂, Mg-Li-TiS₂ and Mg-Na-TiS₂) samples. Peak frequency distribution of pristine TiS₂ (blue), discharged TiS₂ (yellow) and charged TiS₂ (green) corresponding to (**c**) $E_g$, (**d**) $A_{1g}$ and (**e**) Sh modes. Full width half maxima (FWHM) distribution of pristine TiS₂ (blue), discharged TiS₂ (yellow) and charged TiS₂ (green) corresponding to (**f**) $E_g$, (**g**) $A_{1g}$ and (**h**) Sh modes of vibration.

band features (Fig. 5b). The appearance of new or disappearance of existing vibrational modes was not observed. Obviously, no phase change had occurred. However, in the Mg-Na system after discharge, the Raman spectrum of TiS₂ showed significant changes as two new bands centered at ~463 cm⁻¹ and ~490 cm⁻¹ emerged. This supports that TiS₂ undergoes a phase transition during intercalation of the larger Na ions, in good agreement with the bulk XRD analysis. The new Raman band could not be identified however, and would require further investigation. As the Mg-Na system showed severely limited co-intercalation based on the ICP-OES, the focus was directed towards the co-intercalation favoring Mg-Li system.

In Supplementary Fig. 6 a juxtaposition of spectra from dis-/charged electrode surfaces are presented in a head to head fashion. Intercalation of Li and co-intercalation of Mg and Li ions in the interlayer reduced the intensity of the Raman bands, which suggested a lowering of the polarizability of the vibrating sulfur atoms. Hence it could be considered that the intercalated ions induced an electronic structure change. The measured Raman spectra were validated by a thorough statistical analysis of the peak frequencies (see methods for details) and the corresponding full-width-half-maxima (FWHM) of $E_g$, $A_{1g}$ and Sh bands of a variety of cycled samples (Supplementary Fig. 7). The three vibrational modes were traced for the Mg-TiS₂ and the Mg-Li-TiS₂ dis-/charged samples along with partially lithiated (Li-TiS₂-mid) and fully de-/lithiated Li-TiS₂ samples as control or reference systems.

The two out-of-plane modes of vibrations corresponding to the $A_{1g}$ and the Sh showed a similar trend where the wavenumbers (Raman frequencies) increased for the discharged samples (Fig. 5d, e). The fully lithiated TiS₂ (Li-TiS₂ first discharge) blue shifted the $A_{1g}$ mode from 331 cm⁻¹ to 336 cm⁻¹ while the Mg and Li-ion co-intercalation (Mg-Li-TiS₂ first discharge) shifted $A_{1g}$ by an additional ~7 cm⁻¹. The Sh band also showed a blue shift of 21 cm⁻¹ (362 cm⁻¹ to 381 cm⁻¹) post-lithiation, while a stronger blue shift of ~29 cm⁻¹ was observed after the first discharge in the Mg-Li system. A blue shift of $A_{1g}$ band in layered materials has been reported for the case that the number of layers increases[38]. For completeness, we have also provided the frequency distributions together with the outlier points in Supplementary Fig. 8.

This general trend can be ascribed to the relative stiffening of the $A_{1g}$ mode as the interlayer van der Waals interaction increases with intercalation, thus increasing the effective restoring forces acting on the sulfur layers. Furthermore, during intercalation, the interaction is not limited to weak van der Waals forces as intercalated ions induce local charge on the sulfur layers, which in turn inflict a long-range Coulombic interaction that further stiffens the $A_{1g}$ mode[40,41]. The stiffening of out-of-plane $A_{1g}$ and Sh vibrational modes observed in the Mg-Li dual cation co-intercalation was similar to that during the Li ion intercalation, except that a stronger blue shift was observed in the Mg-Li system. This was likely due to the higher charge density of Mg²⁺ which amplified the restoring force of the sulfur atoms in the Mg-Li system compared to the intercalation of only Li ions. After charging, the Sh bands in the Li and Mg-Li systems reverted back to Raman frequencies near pristine TiS₂, which demonstrated reasonable reversibility. However, the corresponding $A_{1g}$ bands showed a red shift with respect to pristine TiS₂ sample. This anomalous behavior can be attributed to the enhancement of interlayer separation at the surface level or exfoliation of TiS₂ to smaller fragments which are insensitive to a bulk structure characterization technique like XRD[40].

The in-plane $E_{1g}$ mode showed a red shift of ~10 cm⁻¹ (214 cm⁻¹ to 204 cm⁻¹) and ~15 cm⁻¹ (214 cm⁻¹ to 199 cm⁻¹) upon lithiation in the Li system and co-intercalation in the Mg-Li system, respectively as shown in Fig. 5c. This contradicted the typical outcome that is expected from the classical van der Waals interactions[38,40]. This divergent behavior between in-plane ($E_g$) and out-of-plane ($A_{1g}$) vibrational modes has been recorded previously in other layered materials, like MoS₂ or GaSe[40,41]. Hence a non-negligible Coulombic interaction between the chalcogen atoms and metallic atoms has been proposed to explain such an observation[41]. This result reinforces our previous assumption of an intercalation induced local charge in the sulfur layer which could have assisted the in-plane vibration and caused the red shift. Fig. 5f–h presents line widths of the Raman modes as a function of dis-/charge of the different systems. It was observed that upon intercalation all the bands showed significant broadening. The implication being that the intercalated carrier ions

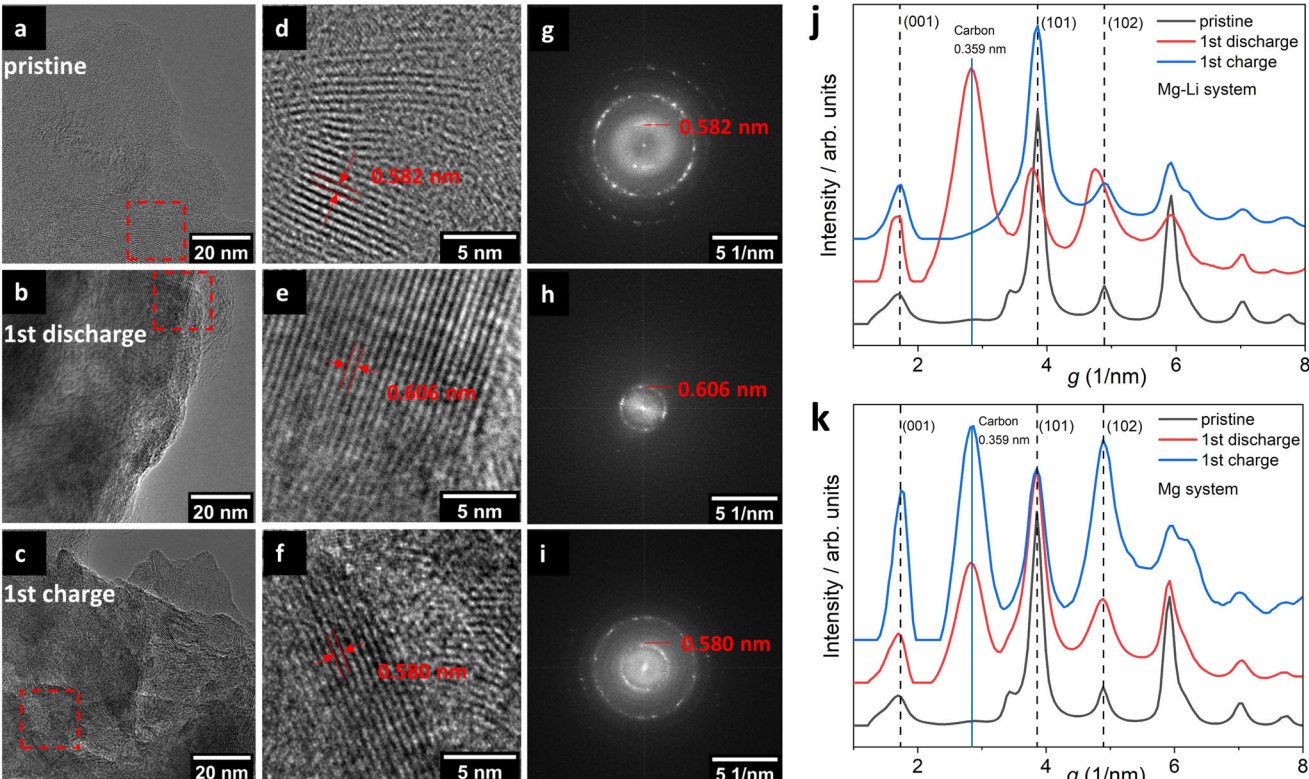

**Fig. 6 | Local structure analysis of TiS₂.** Ex Situ TEM image of TiS2 corresponding to (**a**) pristine, (**b**) discharged and (**c**) charged samples. The corresponding HRTEM images cropped from the area marked with dashed red box correspond to (**d**–**f**). The interlayer distances of marked in the images. **g**–**i** Corresponds to the FFT of TEM images of pristine, discharged and charged samples, respectively. Background subtracted diffraction profiles azimuthally averaged from the corresponding electron diffraction patterns for (**j**) Mg-Li and (**k**) Mg systems. The peak in 0.359 nm is the carbon from the additives.

were distributed in an inhomogeneous manner within the $TiS_2$ interlayer without short-range ordering.

The local structural change of $TiS_2$ in the Mg-Li system was further examined by high-resolution transmission electron microscopy (HRTEM) and electron diffraction. The layered structure of $TiS_2$ was visible in the TEM images shown in Fig. 6a–c. The TEM images of cycled $TiS_2$ (Fig. 6b, c) showed that the layered structure was maintained following de-/intercalation. Different orientations were observed in the HRTEM (Fig. 6d–f) that can be attributed to the polycrystalline nature of $TiS_2$. Interlayer expansion was observed upon co-intercalation of Mg and Li ions as shown in the HRTEM image in Fig. 6e where the interlayer distance increased from 5.82 Å to 6.06 Å upon co-intercalation. After de-intercalation, the interlayer distance contracted back to 5.8 Å as in pristine $TiS_2$, which demonstrated good reversibility. The Fast Fourier Transformation (FFT) patterns (Fig. 6g–i) show ring patterns characteristic for polycrystalline materials. The structural change of $TiS_2$ that was observed in the Mg-Li system was further analyzed with the help of electron diffraction of cycled $TiS_2$. By comparing the diffraction profiles (Fig. 6j, k) azimuthally averaged from the corresponding electron diffraction patterns, it was observed that the (001) reflection shifted from 1.72 nm⁻¹ to 1.65 nm⁻¹ which is equivalent to a d-space change from 5.8 Å to 6.07 Å upon co-intercalation of Mg and Li ions[25]. Additionally, both (101) and (102) reflections shifted towards lower $g$ values in conjunction with the (001) diffraction peak following the expansion of the lattice along $c$ direction. After the corresponding de-intercalation process, the electron diffraction pattern changed back to the pristine pattern (Fig. 6j), further reinforcing that reversible co-intercalation of both Mg and Li ions occurred.

On the contrary, in the Mg system the electron diffraction pattern showed no shift in the position of the reflections (Fig. 6k) as there was

no significant structural change following the limited intercalation of $Mg^{2+}$. Overall, the electron diffraction patterns of both Mg-Li and Mg systems were consistent with the XRD data, which means that the local structural change is consistent with the bulk structural change of $TiS_2$ upon de-/intercalation of the two charge carriers.

## DFT studies

First-principles calculations in the framework of DFT studies were carried out to provide deeper insights into the thermodynamics and kinetics of the cation co-intercalation process. We begin by analyzing the Li, Na, and Mg ion diffusion in the bulk phase of $TiS_2$. The sulfur atoms in $Li_xTiS_2$ and $Mg_xTiS_2$ ($x \leq 1.0$) are stacked in an ABAB sequence, which is the same as the O1-$TiS_2$, crystalized in the P3̄m1 space group (see Fig. 3c). Li and Mg ions can potentially occupy either octahedral or tetrahedral sites in layered O1-$TiS_2$ (denoted as $O_h$ and $T_h$, respectively). The intercalated atoms, however, have the propensity to occupy the $O_h$ sites[34], where each atom is coordinated octahedrally to six S atoms from the two $TiS_6$ octahedra in each of the upper and lower layers of $TiS_2$. The site occupation is more nuanced in thiospinel $Ti_2S_4$ with $Mg^{2+}$ exhibiting mixed occupancy at non-dilute Mg concentrations[42,43]. In the layered $TiS_2$, intercalated atoms typically migrate through the metastable $T_h$ sites when hopping from one $O_h$ site to a neighboring $O_h$ site which normally requires considerable activation energy to overcome the migration barriers. Fig. 7a shows the schematic representation of the diffusion path in O1-$TiS_2$. Our calculated activation energies ($E_a$) were 0.73 eV and 1.28 eV for Li and Mg ions, respectively, as depicted in Fig. 7b (left) and 1.07 eV for Na ions (Supplementary Fig. 9). These values can be used to compute the diffusion coefficient (D) for the ions at room temperature (25 °C) ($k_BT$ = 0.026 eV) using the established Arrhenius expression D ~ exp(-$E_a$/$k_BT$).

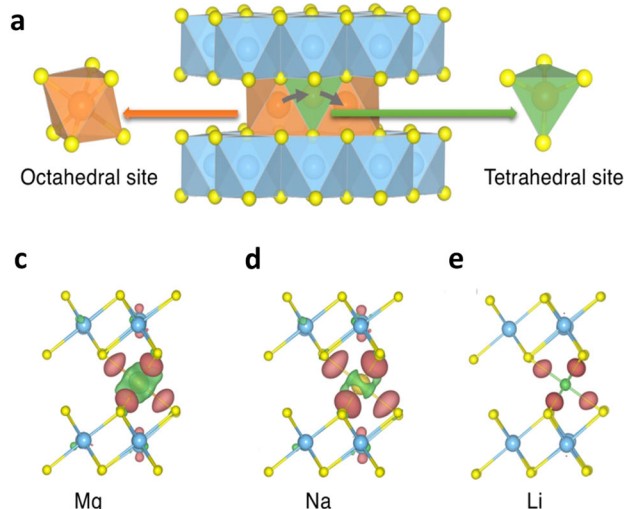

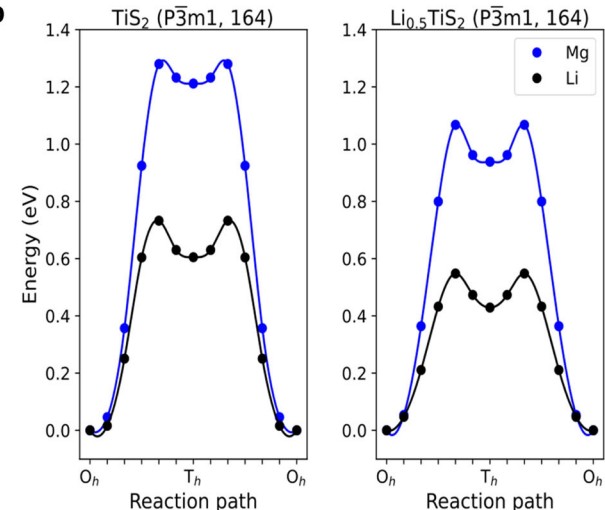

**Fig. 7 | DFT calculations pertaining to the Mg and Mg-Li systems. a** The O1-TiS$_2$ crystal structure with Li and Mg atoms intercalated in octahedral (O$_h$) sites indicated by orange colour. Occupation of the tetrahedral (T$_h$) site is metastable during the hopping process in the diffusion channel between nearby O$_h$ sites, which is indicated by green. **b** Migration barriers of Li and Mg ions in the bulk of TiS$_2$ and Li$_{0.5}$TiS$_2$, respectively. A significant reduction in the height of the activation barrier is observed when Li fills 50% of interlayer sites. **c–e** Show a comparison of the spatial distribution of the difference in charge density (with isovalue of 0.004 eÅ$^{-3}$) for (**c**) Mg, (**d**) Na, and (**e**) Li atom in an O$_h$ site. The brown and green isosurfaces represent the charge accumulation and depletion regions, respectively.

Specifically, the calculated diffusion coefficients of Li and Mg ions at 25 °C are approximately $10^{-13}$ and $10^{-21}$ cm$^2$/s, respectively. This translates to a D of Mg ions that is $10^8$ times lower than that of Li ions which underlined the significant sluggishness of Mg ions. It should be mentioned that the migration barrier calculations were done without taking the temperature parameter into account (i.e. 0 K).

The charge density differences were constructed by subtracting the total electron density of the system with a single intercalated ion from that of the isolated atom and pristine TiS$_2$ without changing the atomic positions. An isovalue of 0.004 eÅ$^{-3}$ was used for all three ions. The distribution of the charge density difference of Li and Na intercalation at equilibrium are comparable, with the exception that the charge depletion zone around Na is more delocalized than that around Li (Fig. 7d, e). In line with earlier observations, the charge rehybridization or electronic redistribution upon Mg insertion appears to be higher than that of Li and Na. The divalent character of Mg ions strongly encourages charge rehybridization (Fig. 7c), explaining why Mg ions move slowly in layered TiS$_2$, highlighting the necessity for new mitigating strategies. In this regard it was observed that the presence of Li in the interstices of TiS$_2$ crystal structure enabled improved solid-state diffusion of Mg. The variations in the energy profiles governing the diffusion of Mg and Li ions within Li$_{0.5}$TiS$_2$ were examined as the interlayer spacing increased from its equilibrium state to a 4% expansion. As illustrated in Fig. 7b (right), we have determined the activation energies (E$_a$) of Li and Mg ions in Li$_{0.5}$TiS$_2$ to be 0.55 and 1.07 eV, respectively. The lowering of the migration barrier for Mg ions amounts to 0.20 eV and for Li ions it is 0.18 eV. This demonstrated a noteworthy reduction in the activation barrier of Mg ions together with its Li counterpart, resulting in a significant enhancement of its diffusion coefficient.

We now shift our attention to the mobility of the Na and Mg ions in the TiS$_2$ crystal structure for the Mg-Na system. Two space groups that were observed experimentally matched well with the Na$_{0.5}$TiS$_2$ and NaTiS$_2$ structures, namely R3m (P3) and R$\bar{3}$m (O3), respectively. In the O3-TiS$_2$ structure, there are O$_h$ and T$_h$ sites for the Na and Mg ions to reside. However, both Na and Mg ions tend to occupy the O$_h$ sites. Compared to the O1 structure, the T$_h$ sites in the O3 phase function as transition states for Na ions due to their larger radii while providing a local minimum for the smaller Mg ions as shown in Fig. 8a. The

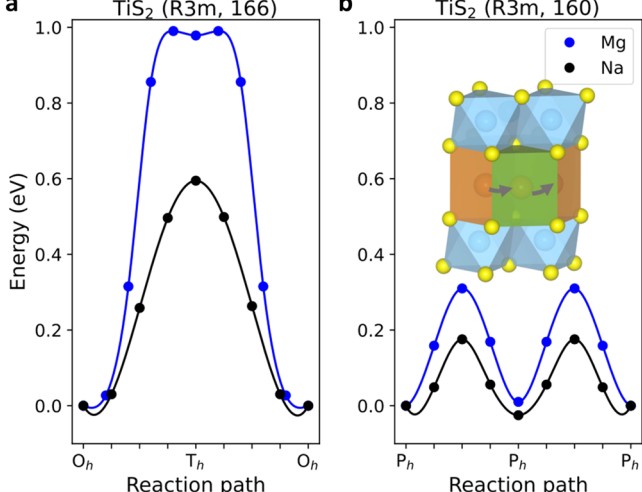

**Fig. 8 | DFT calculations pertaining to the Mg-Na system.** The calculated activation barriers of Na and Mg ion bulk diffusion in (**a**) O3- and (**b**) P3-TiS$_2$. The activation barriers are significantly reduced in the P3 structure compared to the O3 structure due to the rectangular face-sharing transition state.

potential diffusion pathways between the two closest O$_h$ sites going through T$_h$ sites were examined to determine the activation energies. Our computed activation energies were 0.59 and 0.99 eV for Na and Mg ions, respectively (Fig. 8a), which indicated faster diffusion of Mg in the Mg-Na system. For the P3-TiS$_2$ structure, the ABBCCA sulfur stacking and four TiS$_2$ sheets per unit cell have prisms sharing a face with one TiS$_6$ octahedron and three edges with TiS$_6$ octahedra from the next layer, resulting in only one storage site called the prismatic site (see Fig. 4d). Facile migration of Na and Mg ions between two adjacent prismatic sites was obtained with the rectangular shared face as the transition state (Fig. 8b). Our computed activation energies were 0.18 and 0.31 eV for Na and Mg ions, respectively, which again suggested faster mobility. Calculated migration barriers of the O3 and P3 phases however did not align with the experimental finding of suppressed Mg co-intercalation. Therefore, the solid-state diffusion of Mg

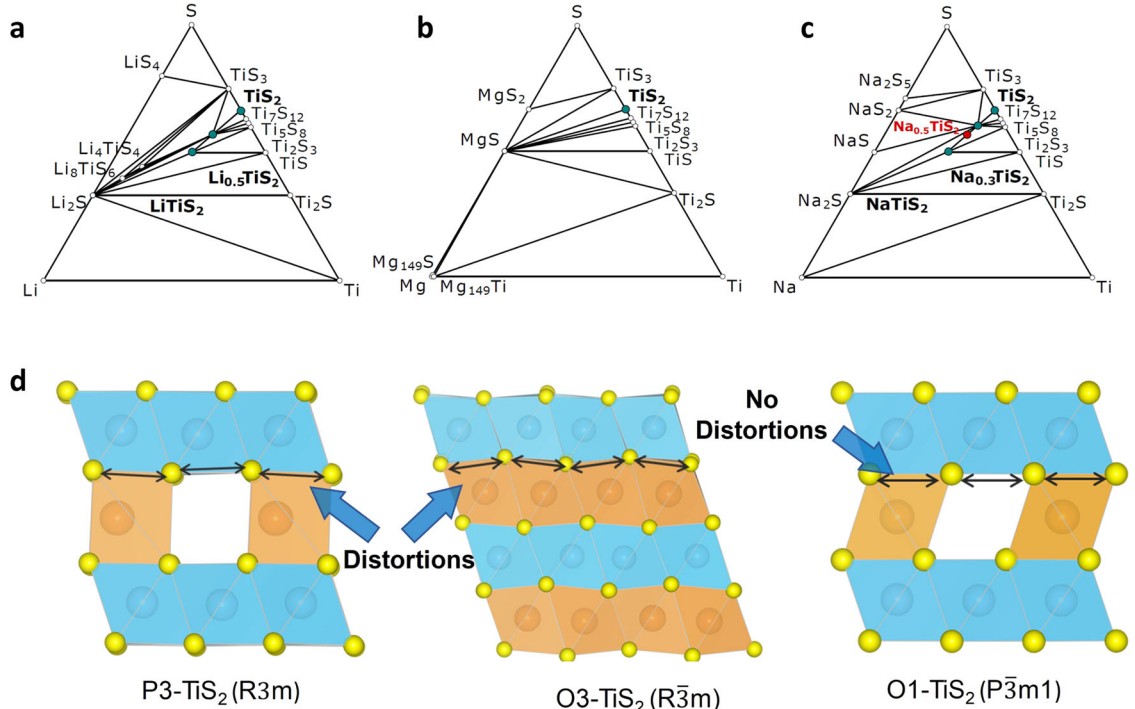

P3-TiS$_2$ (R3m)          O3-TiS$_2$ (R$\bar{3}$m)          O1-TiS$_2$ (P$\bar{3}$m1)

**Fig. 9 | Phase and structural stability of TiS$_2$ and its effect on co-intercalation.** Ternary phase diagrams of the (**a**) Li–Ti–S, (**b**) Mg–Ti–S, (**c**) Na–Ti–S systems. **d** Simulated structures of P3, O3 and O1-TiS$_2$. The labels and points (dark green) correspond to the stable crystalline phases (**a**–**c**) according to the convex hull analysis. The lines highlight the reaction vectors for the various elemental concentrations for the stable phases. The red point in (**c**) corresponds to the metastable half-sodiated P3-TiS$_2$ phase. The effect of the structural stability on interstitial Mg ion occupation is highlighted in the computed structures of P3, O3 and O1-TiS$_2$, where O3 and P3-TiS$_2$ exhibit structural distortions unlike the O1-TiS$_2$.

ions was probably not the determining factor for its limited co-intercalation with Na ions in bulk TiS$_2$.

In order to clarify this anomalous finding, we checked the thermodynamic stability of the different phases of TiS$_2$. Layered sulfides exhibit mainly O1, O3, and P3 structures. As TiS$_2$ slabs glide over each other without breaking S links, transitions between them can happen quickly during de-/intercalation. While TiS$_2$ favors the O1 structure for Li and Mg storage, fully sodiated (-1 Na) TiS$_2$ is stable in the O3 (R$\bar{3}$m) phase, and at intermediate concentrations, the structure exhibits the metastable P3 (R3m) phase[31].

The calculated phase diagrams demonstrate that each ion can lead to a series of phase transformations upon changing its concentration, as shown in Fig. 9. The ternary phase diagrams were constructed by utilizing all DFT computed bulk energies of corresponding materials. No binary phase exists between the metallic phase of the intercalated ions (Li, Mg, and Na) and the titanium phase, which is obviously valid for all of the three phase diagrams. Considering the case of the Li-Ti-S phase diagram (Fig. 9a), we found two stable binary compounds between the lithium and sulfur phases, Li$_2$S and LiS$_4$. On the other hand, multiple stable binary compounds exist between the titanium and sulfur phases, showing a rich transition metal–sulfide chemistry. The ternary phases in this phase diagram exhibited both Li$_{0.5}$TiS$_2$ and LiTiS$_2$, as well as Li$_4$TiS$_4$ and Li$_8$TiS$_6$. The Li intercalated TiS$_2$ showed the same O1 structure for Li-poor (Li$_{0.5}$TiS$_2$) to Li-rich (Li$_8$TiS$_6$) phases. Going from the Li to the Mg system (Fig. 9b), two stable binary compounds, MgS and MgS$_2$, were found between the magnesium and sulfur phases in the phase diagram. No stable ternary phase was calculated in this phase diagram. Therefore, the co-intercalation approach with Li ions could be one way of introducing Mg ions into the TiS$_2$ structure.

The Na system (Fig. 10c) has four binary phases between sodium and sulfur, Na$_2$S, NaS, NaS$_2$ and Na$_2$S$_5$. The titanium and sulfur binary compounds were the same as in Li and Mg phase diagrams. However,

only two phase stable ternary compounds exist, in the form of NaTiS$_2$ and Na$_{0.3}$TiS$_2$ (dark green points) along with the metastable Na$_{0.5}$TiS$_2$ phase (red point). The metastable nature of P3-Na$_{0.5}$TiS$_2$ thermodynamically drives the transition towards phase stable O3-TiS$_2$. In order to clarify the role that the stability of the structure played to inhibit the intercalation of Mg ions in the Mg-Na system, the structural stability of the P3 and O3 phase was calculated upon incorporation of Mg ions. The outcome was that significant distortions were induced in the crystal structure of both P3 and O3-TiS$_2$ (Fig. 9d). Furthermore, it was observed that the accommodation of Mg ions in O3-TiS$_2$ generated relatively stronger distortions compared to the metastable P3-TiS$_2$ (Supplementary Table 3). Thus, the transition of the metastable P3 phase to the thermodynamically stable O3 phase is likely to exacerbate the overall distortions in the crystal lattice, leading to severe structural destabilization. This phenomenon explained well the suppression of Mg$^{2+}$ co-intercalation, as well as the structural degradation observed with cycling (Supplementary Fig. 5). On the flipside, in the Mg-Li system the O1-TiS$_2$ phase is retained when storing the smaller Li ions. Therefore, no distortions were introduced while accommodating Mg ions in the lithiated O1 phase (Fig. 9d) due to the similarity of the ionic radius.

From the computational results, it can be inferred that the diffusion kinetics of Mg ions benefit from the thermodynamic and structural stability of TiS$_2$ as evidenced from the difference observed between the Mg-Li and the Mg-Na systems. This underlines the importance of the ionic radius of the co-intercalating monovalent ion for the strategy to succeed.

## Reaction mechanism of de-/intercalation
The electrochemical reaction mechanism of TiS$_2$ was investigated thoroughly for the co-intercalating Mg-Li system by probing the local electronic structure using ex situ transmission electron microscopy-based electron energy loss spectroscopy (EELS). The high sensitivity of

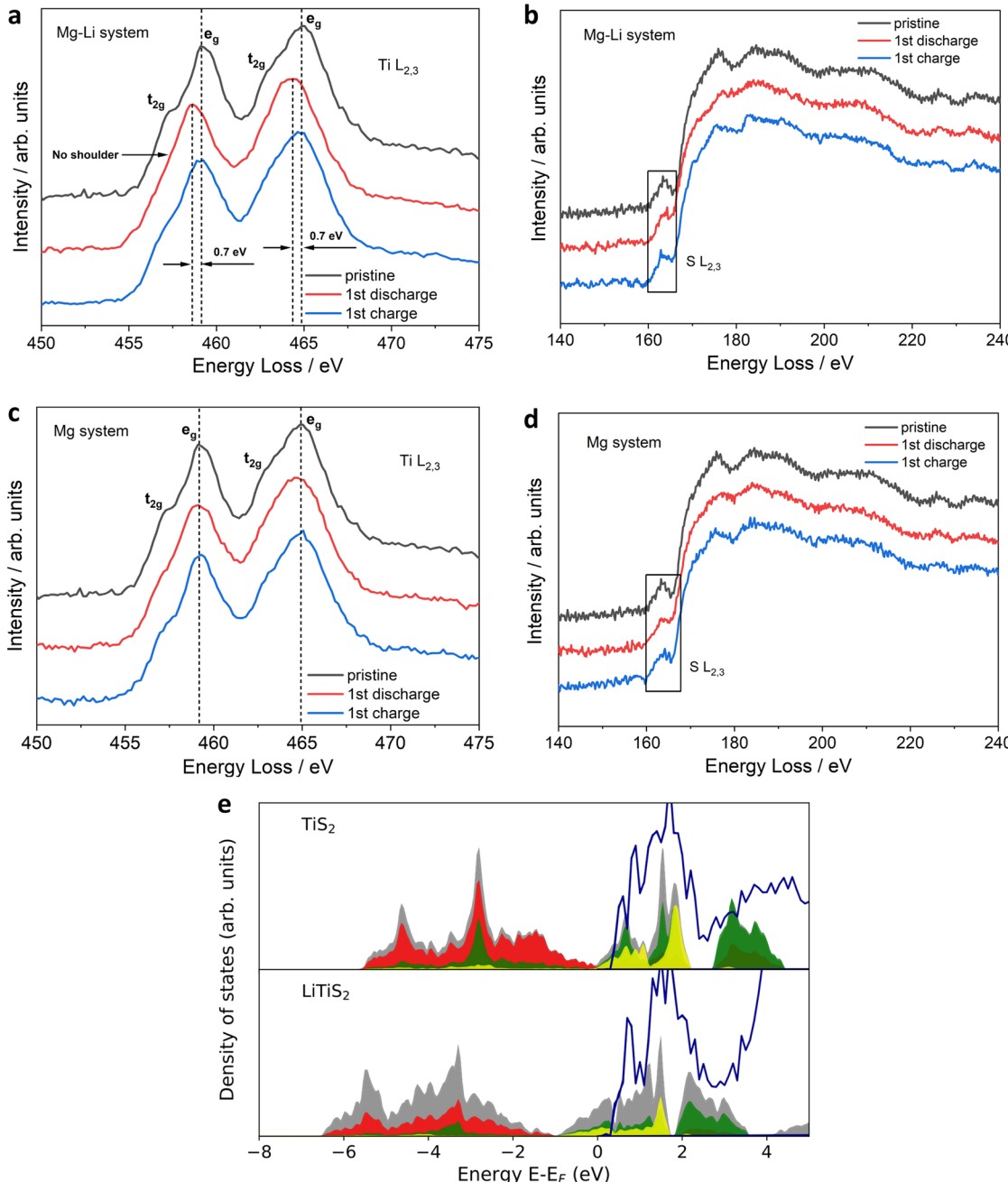

**Fig. 10 | Reaction mechanism investigation by EELS spectra analysis.** EELS spectra of $TiS_2$ in the (**a**, **b**) Mg-Li and (**c**, **d**) Mg systems. The energy range of (**a**, **c**) Ti $L_{2,3}$ edge and (**b**, **d**) S $L_{2,3}$ edge. **e** The calculated density of states of (DOS) of $TiS_2$ (top) and $LiTiS_2$ (bottom) structures. The graphs show the total DOS (grey) and the projected DOS for S-p (red), Ti-$t_{2g}$ ($\sigma$) / Ti-$t_{2g}$ ($\sigma^*$) (green), Ti-$e_g$ ($\sigma$) / Ti-$e_g$ ($\sigma^*$) (yellow) bonding / anti-bonding states. The calculated EELS data is represented by dark blue colour.

EELS for chemical state changes upon redox reaction was exploited to monitor the oxidation state of both Ti and S redox centers. As $TiS_2$ has a covalent character due to the orbital mixing of S 3p and Ti 3d, it has been reported that redox contribution from S is also likely together with Ti[25,44].

In order to clarify the exact nature of the redox reaction, both Ti $L_{2,3}$ and S $L_{2,3}$-ionization edges were analyzed post discharge/charge. A detailed comparison of the Ti $L_{2,3}$ fine structure before and after the first cycle is shown in Fig. 10a. The Ti 3d character of the pristine sample was characterized by the presence of three-fold degenerate $t_{2g}$ states (shoulder peaks) and two-fold degenerate $e_g$ states (peaks at 459.2 eV and 464.9 eV). The crystal field splitting ($t_{2g}$ – $e_g$) is reasonably resolved in $TiS_2$ as can be seen from the clear shoulder peaks at

457.4 eV and 463.1 eV. However, on discharging the Mg-Li system, the shoulder peaks disappeared. This indicates a reduction in the $t_{2g}$ – $e_g$ splitting of the Ti $L_{2,3}$ edge, which is characteristic for the reduction of Ti[45–48]. Furthermore, the Ti $L_{2,3}$ edge was shifted by -0.7 eV to lower energies which confirmed the reduction of the Ti oxidation state. On charging, the Ti $L_{2,3}$ edge shifted back to its previous position and concurrently the shoulder peaks reappeared, which demonstrated a consistent reversible redox process, whereas the S $L_{2,3}$ edge (Fig. 10b) on the other hand showed no significant change in its feature or position upon discharge/charge, which indicated no obvious anionic sulfur redox reaction in $TiS_2$ upon reversible $Mg^{2+}$ and $Li^+$ co-intercalation. The Mg system on the other hand, showed no low energy shift of the Ti $L_{2,3}$ edge after discharge. This is consistent with

the low capacity and incomplete reduction of $TiS_2$ due to the limited intercalation of $Mg^{2+}$ charge carriers. Additionally, the S $L_{2,3}$ edge (Fig. 10d) also showed no clear changes, suggesting no sulfur redox, similar to the Mg-Li system.

Furthermore, the calculated density of states (DOS) alongside calculated EELS data also corroborated the above findings, as illustrated in Fig. 10e. For $TiS_2$, the filled valence band (bonding states, $\sigma$), which extends from −5.5 eV to 0 eV, is predominantly of S-p character with some contribution from Ti-d orbitals. The Ti-d states relevant for the redox properties lie mostly above the Fermi level (anti-bonding states, $\sigma^*$) between 0 and 4 eV. Passing from $TiS_2$ to $LiTiS_2$, the monovalent Li ions are introduced into the system and extra electrons fill up the $\sigma^*$ states. Based on the DOS calculations, the Ti ($\sigma^*$) states were occupied. A detailed look at the spectral region of interest of the calculated EELS showed that the redox activity is related to peaks that lie in the energy window from 0 to 3 eV. The disappearance of the lower peak corresponding to the Ti-$t_{2g}$ ($\sigma^*$) orbital after Li insertion agreed well with the measured EELS data of the Mg-Li system and further verified that Ti is the center of the redox reaction. The bonding/anti-bonding states were further analyzed using the Crystal Orbital Hamiltonian (COHP) analysis as can be seen in Supplementary Fig. 10. It can be observed that upon reduction, the continuum corresponding to the anti-bonding states (blue) near the Fermi level underwent broadening due to the delocalization of the electrons in the empty Ti-d states. Simultaneously, in the bonding continuum (red), which is primarily comprised of the S-p orbitals, the sharp peak vanishes as the bonding weakens likely due to slight lengthening of the Ti−S bond on $Li^+$ intercalation[44]. These observations complement well with the DOS calculations and suggest that Ti is the dominant redox site.

Moreover, charge analysis was also conducted to support the assertion that $TiS_2$ predominantly undergoes redox at the Ti site. It should be noted that from the study done by Zhang et al. and the subsequent Bader charge analysis, it was found that one titanium (Ti) atom and two sulfur (S) atoms gained electrons when lithium ions were intercalated in $TiS_2$[25]. They predicted that both Ti and S contributed to the charge compensation mechanism, with S gaining more electrons than Ti upon intercalation of one $Li^+$ per formula unit. In our study, we found that the PBE (Perdew-Burke-Emzerhof) functional predicts incorrectly the charge compensation process in $TiS_2$ showing a shared mechanism between Ti and S with more contribution from S (Supplementary Table. 4). However, the Ti cations gain more electrons using the more accurate HSE06 hybrid functional[49] and the contribution of the S anions to the charge compensation process becomes relatively small (0.128 for Ti and 0.040 for S), following the charges listed in the VASP (Supplementary Table. 5).

The conclusion that can be derived from the EELS fine structure analysis and the calculated DOS, COHP and charge analysis was that $TiS_2$ did not exhibit any obvious sulfur redox activity and Ti is the dominant redox active site when charge carriers are intercalated. The redox mechanism was more clearly observed for the co-intercalating Mg-Li system compared to the Mg system as a greater amount of charge was transferred to the Ti redox center due to the co-intercalation of two charge carriers.

## Discussion

We have demonstrated herein a novel approach to offset the poor redox activity of Mg ions in model $TiS_2$ cathode by enabling dual cation co-intercalation with faster Li ions by using a fluorinated alkoxyborate-based dual salt electrolyte. The concept was then extended to an analogous system by incorporating Na ions instead of Li and the performance was similar, delivering an initial discharge capacity close to the theoretical value. Previously, dual salt electrolytes have been explored to mitigate the high migration barrier and associated kinetically sluggish transportation of $Mg^{2+}$. However, the widely reported dual salt electrolytes are either corrosive due to the presence of

chlorine, like APC−LiCl/LiBF₄, where APC stands for All Phenyl Complex, or are highly reductive, in the case of borohydrides $(Mg(BH_4)_2−NaBH_4))$[29,50]. Both these classes of electrolyte have limited anodic stability, which restricts them from high voltage applications. Yagi et al. reported self-discharge and spurious side reactions involving an APC-LiBF₄ / THF (tetrahydrofuran) dual salt and LiFePO₄ cathode during resting and charging, respectively[51]. Furthermore, borohydride based $Mg(BH_4)_2−NaBH_4$ / DGM (diglyme) has been shown to be chemically reactive by irreversibly modifying the pristine $TiS_2$ cathode[29] after the first discharge and lowered the anodic stability when the concentration of $Mg(BH_4)_2$ was increased. In addition, there is strong association between anion and cation in both Cl-based and borohydride-based electrolytes, generating a high energy barrier for dissociation at cathode-electrolyte interface. The respective monovalent cation species (such as $MgCl^+$) might even intercalate into the cathode as a whole, which makes it more complicated for exploring the cation co-intercalation strategy. In comparison, the dual salt electrolytes used in this work have superior anodic stability due to the strong C-F bond in $[B(hfip_4)]^-$ which render it as an ideal electrolyte system for exploring the cation co-intercalation strategy[52].

Typically, the dual salt electrolyte approach has been tuned towards designing hybrid battery systems, where the positive electrode only accommodates the faster $Li^+/Na^+$ (discharge) while the slower $Mg^{2+}$ plates on the metal negative electrode (charge). Effectively, the cathode operates as a $Li^+/Na^+$ pass filter[50]. This strategy has been reported to be a viable way to circumvent the problematic kinetics of $Mg^{2+}$. However, as the charge carriers (all $Mg^{2+}$ after discharge and all $Li^+$ after charge) are stored in the electrolyte solution, large amount of solvent is required, which consequently lowers the energy density. Not to mention, the cell design will need to be altered to safely hold larger quantities of electrolyte. On the contrary, dual cation co-intercalation enabled by a dual salt electrolyte is based on the 'rocking chair' principle[22]. The advantage of this approach is that the charge carriers are accommodated in the electrodes and not in the electrolyte, thus ensuring higher overall energy density.

Co-intercalation approaches have been investigated as an alternative counter measure to combat the sluggishness of $Mg^{2+}$. Water solvated $Mg^{2+}$ was first studied by Song et al. to shield the charge carrier and decrease its polarization[53]. Although the capacity was much improved, the presence of water in the electrolyte passivated the Mg anode[54]. Along similar lines, $V_2O_5$ xerogels with hydrated interlayers screened the intercalated $Mg^{2+}$ and reduced the polarization. However, it was later reported that the water molecules get extracted on charging along with $Mg^{2+}$, causing the $V_2O_5$ structure to collapse. A more effective solvent co-intercalation method was reported by Li et al. with DME solvated $[Mg \cdot 3DME]^{2+}$ exhibiting fast kinetics in layered $MoS_2$ by shielding the high charge density of $Mg^{2+}$[20]. A study done by Yoo et al. explored an alternative co-intercalation strategy and demonstrated fast kinetics in $TiS_2$ by intercalating monovalent $[MgCl]^+$. This can be interpreted as the co-intercalation of $Mg^{2+}$ and $Cl^-$ ions[55]. Nevertheless, the intercalation of such a bulky specie, either $[Mg(solvent)_x]^{2+}$ or $[MgCl]^+$, could cause steric hindrance and hence require artificial modification of the host structure. Furthermore, cation-solvent co-intercalation or cation-anion co-intercalation requires large amount of electrolyte as reservoir of solvents or anions which in turn will affect the energy density.

The co-intercalation concept reported in this work corresponds to the simultaneous accommodation of two cations in the host crystal structure without the need for any artificial structural modification of the cathode due to the much smaller size of cationic charge carriers. Thus, ensuring greater structural integrity and stability. Additionally, dual cation co-intercalation strategy executed by the dual salt electrolyte approach also has the possibility to lower the de-solvation penalty at the cathode-electrolyte interface, thus enhancing the electrochemical performance[56]. Furthermore, compared to the other co-

intercalation methods, the electrolyte amount is not a rate limiting parameter for achieving higher energy densities. Finally, both cations are active charge carriers and enable obtaining high specific capacities.

Investigation of the dual cation co-intercalation strategy requires a thorough examination of the capacity contributions from each charge carrier. This was conducted through elemental analysis, which revealed an interesting dichotomy underlining the influence of the ionic radius of the cations. While the Li ions enabled co-intercalation of $Mg^{2+}$, the influence of the bigger Na ions was detrimental. Examining the reason behind the divergent behavior between two apparently similar systems highlighted the impact of phase transformation on the ability of $TiS_2$ to accommodate two different charge carriers with different valencies. The Mg-Li system exhibited no phase change of O1-$TiS_2$ upon co-intercalation of Mg and Li ions. The thermodynamic and structural stability of a variety of lithiated ternary $Li_xTiS_2$ compositions along with interlayer expansion enabled improved solid-state diffusion of $Mg^{2+}$ with the migration barrier being reduced by more than 0.2 eV. On the other hand, $TiS_2$ underwent a phase change to a mixture of metastable P3 and O3 phases upon the first full discharge in the Mg-Na system. Despite greater interlayer expansion, the metastability of P3 phase and the unfavorable energetics of the prismatic site suppressed the co-intercalation of the smaller and densely charged $Mg^{2+}$. In fact, calculations and experiments showed that the incorporation of $Mg^{2+}$ in P3 and O3 structures led to severe structural distortions which clarified the reason behind the low Mg concentration.

A detailed mechanistic study of the Mg-Li system clarified the charge storage mechanism in $TiS_2$. Post-mortem ex situ analysis of $TiS_2$ showed that the co-intercalation was reasonably reversible with interlayer expansion/contraction being observed after full dis-/charge. Raman spectroscopy showed the interaction of the intercalated ions with the surrounding sulfur atoms. Comparing the vibrational modes of the Mg-Li system with the half-discharged and fully discharge Li systems further highlighted that Mg and Li ions were co-intercalated into the structure. This was clarified as a stronger interaction was observed with the surrounding sulfur atoms in the Mg-Li system, comprising of 0.35 Li together with 0.24 Mg, in comparison to the half lithiated Li system (half-discharged). Moreover, when compared against the fully lithiated Li system, the Mg-Li system still exhibited a slightly stronger blue shift in the out-of-plane frequencies by 8 cm$^{-1}$ likely due to stronger Coulombic interaction due to the presence of divalent $Mg^{2+}$. The above specified observations lend credence towards improved storage of $Mg^{2+}$ in tandem with $Li^+$ in the Mg-Li system, whereas the single ion Mg system suffered to sufficiently intercalate $Mg^{2+}$ and showed no obvious change in the Raman modes due to dilute Mg concentration in the $TiS_2$. Furthermore, the redox mechanism investigation clarified that Ti is the dominant redox center and no discernible anionic redox is observed in $TiS_2$.

Overall, the co-intercalation method proved to be a viable strategy to mitigate the poor solid-state diffusion of divalent Mg ions which opens an alternative pathway to improve the performance of RMBs provided the right accompanying monovalent ion is employed. The conclusions drawn from the model systems provide guidelines to further explore co-intercalation chemistries, especially high-voltage cathode materials by designing suitable dual cation systems.

## Methods
### Electrolyte synthesis
Li[B(hfip)$_4$]: LiBH$_4$ powder (0.32 g, 14.8 mmol) was dissolved in DME (25 ml) in a Schlenk flask. To this solution, 4.1 equivalents of hexafluoroisopropanol (hfip) amounting to 10.3 g (6.4 ml, 60.7 mmol) was dropwise added while stirring. After stirring for 1 h, the flask was equipped with a Dimroth condenser and refluxed for 2 h at 85 °C under argon atmosphere. Afterwards, following cooling down the solvent was removed using a vacuum pump equipped with cooling traps. The resulting solid residue was further dried at gradually increasing temperatures from 30 °C to 60 °C under vacuum until a pressure of 0.1 Pa was achieved.

Na[B(hfip$_4$)]: 0.56 g of NaBH$_4$ (14.8 mmol) was dissolved in DME (28 ml) in a 100 ml Schlenk flask. 4.1 equivalents of hexafluoroisopropanol (10.3 g, 6.4 ml, 60.7 mmol) was slowly added in a dropwise manner while stirring. The mixture was refluxed at 85 °C for 2 h under argon atmosphere. When the mixture cooled down, the solvent was removed using a vacuum pump. The solid residue was further dried under vacuum at gradually increasing temperatures from 30 °C to 60 °C until 0.1 Pa was achieved.

Mg[B(hfip$_4$)]$_2$: Mg(BH$_4$)$_2$ powder (0.80 g, 14.8 mmol) was first dissolved into 50 ml DME in a Schlenk flask. 20.40 g (12.8 ml, 121.4 mmol) or 8.2 equivalents of hexfluoroisopropanol was added dropwise into the solution while being stirred. After stirring for 1 h, the flask was connected with a Dimroth condenser and refluxed at 85 °C under argon for 2 h. A nearly colourless clear solution was formed. After cooling down, the solvent was removed with a vacuum pump. The resulting solid was further dried at gradually elevated temperatures from 30 to 60 °C by vacuum[52].

### Preparation of electrolytes
All electrolyte salts were dissolved in 1,2 dimethoxyethane (DME). The solvent was dried with 3 Å molecular sieves for 72 h before usage and the water content observed via Karl Fischer titration was 1 ppm. [0.3 M Mg[B(hfip)$_4$]$_2$ – 0.15 M Li[B(hfip)$_4$] / DME] dual salt electrolyte was prepared by adding 2478 mg Mg[B(hfip)$_4$]$_2$ and 616 mg Li[B(hfip)$_4$] to DME in a 5 mL volumetric flask. [0.3 M Mg[B(hfip)$_4$]$_2$ – 0.15 M Na[B(hfip)$_4$] / DME] dual salt electrolyte was prepared by adding 991.5 mg Mg[B(hfip)$_4$]$_2$ and 251.1 mg Na[B(hfip)$_4$] to DME in a 2 mL volumetric flask. The single salt electrolyte solutions, 0.3 M Mg[B(hfip)$_4$]$_2$ / DME, 0.15 M Li[B(hfip)$_4$] / DME and 0.15 M Na[B(hfip)$_4$] / DME, were prepared by adding 991.5 mg, 246.3 mg and 251.1 mg of corresponding salts to DME in 2 mL volumetric flasks, respectively. All electrolyte solutions were shaken and after waiting for 1 day, the supernatant was used. It should be noted that the dual salt electrolytes still had some residues, hence the concentrations (0.3 and 0.15 M) are not exact and the molarity values have been used for convenience.

### Electrochemical measurement
Electrochemical measurements were carried out using two different configurations. Swagelok cell configuration was used to carry out galvanostatic measurements while PAT-Cell configuration from EL-CELL was used to conduct CV measurements. $TiS_2$ (99.9%, 200 mesh) was purchased from Aldrich and ball milled at 300 rpm with a milling time of 5 min followed by 10 min of rest, repeated for 16 times. The slurry was prepared by mixing 70% $TiS_2$ as the active material with 20% Super P carbon black (Timcal) and 10% Polyvinylidene difluoride (PVDF) from Solef in N-methyl-2-pyrrolidone (NMP). The slurry was then coated on 11.8 mm polished stainless steel current collectors. The areal active mass loading ranged between 0.8 and 1.2 mg cm$^{-2}$. The cathodes were dried under vacuum at 80 °C for 15 h. The dual salt electrolyte solutions used in this work are [0.3 M Mg[B(hfip)$_4$]$_2$ – 0.15 M Li[B(hfip)$_4$]] / DME and [0.3 M Mg[B(hfip)$_4$]$_2$ – 0.15 M Na[B(hfip)$_4$]] / DME. The magnesium tetrakis (hexafluoroisopropyloxy) borate Mg[B(hfip)$_4$]$_2$ was synthesized by reacting MgBH$_4$ and hexafluoroisopropanol in DME in a one-pot reaction following our previous work[52]. The Li[B(hfip)$_4$] and Na[B(hfip)$_4$] electrolytes were synthesized by following the same reaction steps, using LiBH$_4$ and NaBH$_4$ as precursors, respectively, as explained in the electrolyte synthesis section. The Swagelok cells were assembled in an Ar-filled glovebox with freshly polished 11 mm Mg disks (0.1 mm thick from Gelon LIB group) used as the anode and two 12 mm diameter glass fibre GF/C as the separator soaked with 80 µL electrolyte. Other electrolytes that were

used in this work are 0.15 M Li[B(hfip)$_4$] / DME and 0.15 M Na[B(hfip)$_4$] / DME. For the RMB three-electrode configuration, we incorporated polished Mg foils of 14 mm diameter (0.1 mm thick) as the counter electrode while a Mg ring (EL-CELL) was used as the reference (Mg$_{ref}$). Two 21 mm diameter GF/C separators soaked in 250 μL electrolyte were used. The three-electrode configuration of the lithium half-cell was constructed by incorporating polished 13 mm Li foil (0.75 mm thick) as the counter electrode with Li ring (EL-CELL) as the reference and a single GF/C separator soaked in 100 μL electrolyte. For all, three-electrode CV measurements, 11.8 mm TiS$_2$ electrodes were used as the working electrode. The BCS 805 battery cycler from Bio-Logic was used for the two-electrode galvanostatic measurements. The three-electrode CV measurments were carried out with the SP150 single channel potentiostat from Bio-Logic SAS. All electrochemical measurements were carried out at 25 °C in a controlled climate chamber.

## Characterization

XRD was performed using a STOE-STADI P powder diffractometer operated in the Debye- Scherrer mode by applying Ag-Kα$_1$ radiation (λ = 0.559407 Å) with a step size of 2.04° and a step time of 120 s. Elemental analysis of the samples were carried out with ICP-OES in a Spectro Acros-SOP system. Ex situ Raman spectroscopy was done using an inVia™ confocal Raman microscope from Renishaw. The Raman spectra were collected over a single 30 μm × 30 μm area at spatial resolution of 1.0 μm per pixel. Laser power at the sample was 2.5 mW at 633 nm wavelength with a 30 s exposure time. A Leica™ 50× long working distance, 0.5 N.A. microscope objective was used. Parameters of the laser beam were carefully selected to obtain the maximal signal to noise ratio without sample heating which mask the native state of the electrode surface. The cosmic rays and the background of all the spectra were then removed using Renishaw WiRE™ 4.0 software. The modified Raman spectra were deconvoluted in individual spectral components using Lorentzian line-shape function in MATLAB platform to designate the positions, full widths at half maximum (FWHM), and intensities of Raman modes (see the Supporting Information for details). Transmission Electron Microscopy (TEM) characterization was performed on a double aberration-corrected microscope ThemisZ (ThermoFisher Scientific) at an acceleration voltage of 300 kV. The TEM is equipped with a high angle annular dark-field (HAADF) detector for scanning transmission electron microscopy (STEM) and a Super-X energy-dispersive X-ray spectroscopy (EDX) detector to acquire EDX elemental maps. EELS data were acquired with an energy resolution of ≈1 eV, estimated from the FWHM of the zero-loss peak using a Gatan image filter with K3 camera (Gatan Inc.). For all ex situ measurements including ICP-OES, XRD, Raman spectroscopy and STEM, the samples were collected from cathodes of cells at specified dis-/charge states. The cells were disassembled in an Ar atmosphere glovebox and the cathodes were washed several times with DME. Afterwards they were dried at 60 °C for 12 h under vacuum before the samples were collected for corresponding ex situ analysis. The ex situ XRD was performed under Ar atmosphere by sealing the powder sample scratched off the cathode in a 0.6 mm wide glass capillary. For ICP-OES, 3–4 mg of cathode material (including carbon super P and PVDF) was dissolved in aqua regia (HNO$_3$:HCl in a 3:1 ratio) to prepare each sample. The resulting solution was diluted with deionized water, depending on the detection and calibration regime of the element of interest. Filtration was necessary in all cases as carbon super P did not dissolve in the solution. The Raman samples were prepared by sealing the powder cathode samples between two glass slides with UHV epoxy in an Ar glovebox. The TEM specimens were also prepared inside the glovebox by scratching the electrode material with a lacey carbon-coated copper grid. Afterwards, the grid was mounted into a Gatan 648 vacuum transfer holder, which can be used to transfer the specimen into the TEM under Ar protection.

## Statistical information

A collection of 360 spectra were collected from the electrode surface, which contains Raman fingerprint of both active materials and additive conducting carbon. Spectrum from each pixel was crosschecked and only 90 spectra were selected which contained less carbon signature. The selected Raman spectra are deconvoluted into three different spectral portions $E_g$, $A_{1g}$ and 'Sh' by using three Lorentzian line shape functions in a MATLAB script. The fitting parameters e.g., peak frequency or FWHM of a band and their relative occurring frequency is plotted in x and y axis respectively in a histogram plot. The statistical occurrence frequency plots are the fitted under normal distribution (Supplementary Fig. 7). The normal distributive nature of the histogram establishes a confidence on the sampling frequency which is 90 (90 selected spectra among 360 on the electrode surface).

The comparisons among peak frequencies and respective FWHM from the different charge discharge states are presented in a box chart form. A box chart plot displays the five-number summary of a set of data. The five-number summaries are first quartile-third quartile (25–75%) enclosing box, the whisker selecting points in the ± 1.5 times of the quartile (IQR), the median line, the mean point and outliers (Supplementary Fig. 9). This also validates a statistical viewpoint which is essential in analyzing a relatively inhomogeneous surface.

## DFT calculations

Density functional theory (DFT) was employed based on the generalized gradient approximation using the PBE[57] functional and the projected augmented[58] wave (PAW) method[59,60] to describe ion-electron interactions as implemented in the Vienna ab initio simulation package (VASP)[61]. The underlying structural optimizations included the third-generation (D3) semi-empirical van der Waals corrections proposed by Grimme[62]. The plane-wave cutoff energy was set to 520 eV, and the Brillouin zone was represented by Monkhorst-Pack (MP) k-point meshes of $2 \times 2 \times 2$[63]. Activation barriers and minimum energy pathways for the charge carriers hopping were obtained using the climbing-image nudged elastic band method (cNEB)[64,65]. The diffusion path was first constructed by linear interpolation of the atomic coordinates in the initial and final states and then relaxed until the forces on all atoms were smaller than 0.05 eV Å$^{-1}$. Large supercells were chosen to ensure that each ion is isolated from its periodic images (images of ions are no less than 6 Å apart). The configurations for the construction of the phase diagrams are taken from the Materials Project (MP) database[66].

## Data availability

The experimental and theoretical data generated in this study have been deposited in the Zenodo database under accession code: https://doi.org/10.5281/zenodo.8355348.

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

## Acknowledgements
The authors acknowledge the funding from the German Research Foundation (DFG) under Project ID 390874152 (POLiS Cluster of Excellence). This work contributes to the research performed at CELEST (Center for Electrochemical Energy Storage Ulm-Karlsruhe). Z.L. acknowledge the funding by the National Natural Science Foundation of China with grant No 52002350. The authors also acknowledge the computational time provided by the state of Baden-Württemberg through bwHPC and the German Research Foundation (DFG) through grant no INST 40/575-1 FUGG (JUSTUS 2 cluster). The authors would further like to acknowledge the ICP-OES operator, Jason Lelovas, for conducting the ICP-OES measurements.

## Author contributions
A.R. prepared the electrolytes, fabricated the electrodes and conducted all electrochemical tests. A.R., M.S. also performed all ex situ XRD measurements and the associated analyses and A.G. did the DFT calculations and contributed to the theoretical part of the manuscript. S.D. was responsible for the ex situ Raman spectroscopy and related analysis. Y.T. and C.K. conducted the STEM and EELS measurements and all related analyses. A.R. wrote the manuscript with contributions from M.S., S.D., Y.T., and Z.L. The project was conceptualized and coordinated by M.F., Z.Z.-K., and Z.L. All authors have given approval to the final version of the manuscript.

## Funding

## Competing interests
The authors declare no competing interests.
