## [Peer Review File · Nature Communications]

REVIEWER COMMENTS

Reviewer #1 (Remarks to the Author):

The work submitted to Nature Communications is very interesting, and the authors made a great job to progress in the field of dual-ion batteries. I think that the manuscript could be recommended for its publication after revision.

Dual Li-Mg and Na-Mg electrolytes have been previously reported in the literature. The progress in this field (electrolyte composition) claimed in the work should be further explained and compared with the literature. The electrolyte solution is mentioned several times in the manuscript. Please, explain the advantage of the employed electrolyte solutions compared to others. The desolvation of cations in the electrolyte solutions before intercalation can be different for Li, Na and Mg. Please, comment if the desolvation process influences electrochemistry.

In Fig. 2c, the electrochemical behavior of layered TiS₂ in Mg-system, is very different compared to the results shown in reference number 25 [Sun (2016)]. The electrochemistry of TiS₂/Mg reported by Linda Nazar's group in reference 25 is better and the voltage-profile is different. This should be explained in the manuscript.

In Fig. 2b, the electrochemistry in Na-Mg-system should be further compared with the literature. For example, this reference: Bian et al. *J. Mater. Chem. A*, 2017,5, 600-608.
<https://doi.org/10.1039/C6TA08505A>.

Lines 427-428 say that Li and Mg ions can potentially occupy either octahedral or tetrahedral sites in O1-TiS₂. Accommodation of cations in tetrahedral sites of spinel TiS₂ has been reported by other authors. For example, see these references, and comment on them:

- *Chem. Mater.* 2018, 30, 7, 2436. <https://doi.org/10.1021/acs.chemmater.8b00552>.

- *Energy Environ. Sci.* 2016, 9, 2273. <https://doi.org/10.1039/C6EE00724D>

In the XRD patterns (Fig. 3), after looking at the low-intensity reflections at around 20 ° two-theta, it seems to me that these two reflections are reversibly modified in the discharge/charge of the Mg-Li system (Fig. 3a), but this process is irreversible in the Mg-system (Fig. 3b). Please, comment on that.

The theoretically calculated activation energy for Li and Mg migration is lower in $\text{Li}_{0.5}\text{TiS}_2$ (Fig. 8b) than TiS_2 . I wonder if the energy reference (arbitrarily 0.0 eV in Fig. 8b) of $\text{Li}_{0.5}\text{TiS}_2$ is not higher than the energy reference of TiS_2 , and this may be the reason for the apparent lower activation energy. Thus, $\text{Li}_{0.5}\text{TiS}_2$ could be like a sort of energized or entatic state as was recently proposed for certain electrodes [Pérez-Vicente. *PhysChemChemPhys* 25 (2023) 15600. DOI: 10.1039/d2cp06044b], due to cation-cation repulsions, with a smaller height of the hopping, compared to TiS_2 . I would suggest calculating the energy of Mg site (octahedral) for TiS_2 and $\text{Li}_{0.5}\text{TiS}_2$.

Reviewer #2 (Remarks to the Author):

This study aimed to overcome the limited redox activity of Mg ions in a TiS_2 cathode by introducing dual cation co-intercalation with faster Li ions, facilitated by a fluorinated alkoxyborate-based dual salt electrolyte. The concept was extended to an analogous system with Na ions instead of Li, yielding similar performance and achieving an initial discharge capacity close to the theoretical value. Through a combination of experimental and computational techniques, detailed explanations were provided for the divergent behavior observed between these two seemingly similar systems, including the charge storage mechanism and redox mechanism for the model system.

This research may have considerable implications for the advancement of high-voltage cathode materials. Therefore I would like to recommend this work, with the condition that the authors address the following comments in their revision:

1. In the abstract, the objective of the study was stated as the enhancement of Mg^{2+} mobility. However, this paper does not present any electrochemical performances, including rate performances.
2. In the dual-ion systems, the TiS_2 showed capacity fading in the initial cycles. In the extended cycles, the CE was around 90%. What are the possible reasons? In addition, why CE is greater than 100% in Figure S2?
3. In the Mg-Na system, the Mg intercalation was impeded. Note, the TiS_2 was deeply discharged, with the possibility to trigger conversion reaction. Such severe structural change might also be detrimental to further intercalation. The authors should clarify this point.
4. What are the difference or advantages of the cation co-intercalation concept over other co-intercalation approaches reported before? It's recommended to discuss it as perspective or outlook.

5. The authors claimed that "activation energies (E_a) were 0.60 and 1.28 eV for Li and Mg ions, respectively, as depicted in Fig. 8b (left)". However, this claim is inconsistent with the diffusion barrier shown in Fig. 8b, where the diffusion barrier for Li is approximately 0.70 eV. This discrepancy requires clarification. Furthermore, the authors estimated the diffusion coefficient of Mg ions to be 107 orders lower than that of Li based on DFT results. The method used for this estimation should also be clarified.
6. The authors compared the charge difference of Mg, Li, and Na in Figure 8. To ensure a fair comparison at the same level, the isovalues for each of them need to be added. Additionally, the authors could strengthen their claim by conducting a charge analysis, which would provide a quantitative comparison of the ions.
7. The calculated energy barrier for Li intercalation in TiS_2 was ~ 600 meV, which is close to the threshold for intercalation chemistry (625 meV). Experimentally, the kinetics of Li^+ in TiS_2 was fairly good. What made the discrepancy?
8. As the pristine TiS_2 was with a O1 phase, the activation energy for Na^+ migration in O1- TiS_2 should be added.
9. The analysis of the phase diagram presented in the text is not easily comprehensible, particularly for the sodium system. Therefore, it is advisable to rephrase the paragraph in a more comprehensive manner, ensuring clarity and understanding.
10. When examining the redox activity of TiS_2 , relying solely on the calculated DOS may not be sufficient to unveil the bonding/anti-bonding states. Additional methods such as COHP analysis should be incorporated to provide a more comprehensive understanding. Furthermore, it is suggested that charge transfer analysis be included to support the assertion that TiS_2 does not exhibit pronounced sulfur redox activity but rather demonstrates Ti as the primary redox active site when charge carriers are intercalated.

Response to the REFEREE REPORTS

The reports of the reviewer are copied in *italics*.

Reviewer #1

The work submitted to Nature Communications is very interesting, and the authors made a great job to progress in the field of dual-ion batteries. I think that the manuscript could be recommended for its publication after revision.

Response: We appreciate that the reviewer considers our paper to be “very interesting and the authors made a great job to progress in the field of dual-ion batteries”. We thank the reviewer for the insightful comments. We have thoroughly reviewed and considered all the concerns raised by the reviewer. Subsequently, we have made appropriate revisions to the paper, which are elaborated upon in the following sections.

Dual Li-Mg and Na-Mg electrolytes have been previously reported in the literature. The progress in this field (electrolyte composition) claimed in the work should be further explained and compared with the literature. The electrolyte solution is mentioned several times in the manuscript. Please, explain the advantage of the employed electrolyte solutions compared to others.

Response: We thank the reviewer for making this remark and agree that some clarification could help. Primarily, the advantage of the dual-salt electrolyte used in this work is that the design of the alkoxyborate $[B(hfip)_4]^-$ anion provides a high anodic stability due to the strong C-F bonds (3.5 vs Mg on Pt and 4.3 V vs Mg on stainless steel) (<https://doi.org/10.1039/C7TA02237A>). In addition, the electrolyte is chlorine free which makes it non-corrosive to the non-noble metallic parts of the cell. Furthermore, the weakly coordinated anion ensures high degree of dissociation of the electrolyte compounds and thereby fast ionic conductivity.

Majority of the reported dual-salts are either based on the chlorinated APC (All Phenyl Complex: $PhMgCl-AlCl_3$) electrolyte (e.g. [APC-LiBF₄](https://doi.org/10.1039/C3TA13668J), [APC-LiCl in tetrahydrofuran](https://doi.org/10.1002/aenm.201401507), <https://doi.org/10.1039/C3TA13668J> <https://doi.org/10.1002/aenm.201401507>) or electrolytes which belongs to the class of highly reductive borohydrides (e.g. [Mg\(BH₄\)-Na\(BH₄\) in tetrahydrofuran or 1,2 dimethoxyethane THF/DME](https://doi.org/10.1039/C6TA08505A), <https://doi.org/10.1039/C6TA08505A>). Both these classes of electrolytes suffer from low anodic stability limiting their application to low voltage cathodes. In addition, there is strong association between anion and cation in both Cl-based electrolyte and the borohydride electrolyte, generating a high energy barrier for dissociation at cathode-electrolyte interfaces. The respective monovalent cation species (such as $MgCl^+$) might even intercalate into the cathode as a whole (<https://doi.org/10.1038/s41467-017-00431-9>), which makes it more complicated for exploring the cation cointercalation strategy.

Following the remark of the reviewer, we have compared this work with other reported works based on dual-salt electrolytes and have accordingly made necessary changes to the manuscript under the ‘Discussion’ section on pages **25, 26** as follows:

“Previously, dual salt electrolytes have been explored to mitigate the high migration barrier and associated kinetically sluggish transportation of Mg^{2+} . However, the widely reported dual salt electrolytes are either corrosive due to presence of chlorine (APC–LiCl/LiBF₄) or are highly reductive, in the case of borohydrides ($\text{Mg}(\text{BH}_4)_2\text{–NaBH}_4$).^{1,2} Both these classes of electrolyte have limited anodic stability, which restricts them from high voltage applications. Yagi *et al.* reported self-discharge and spurious side reactions involving APC–LiBF₄ / THF dual salt and LiFePO₄ cathode during resting and charging, respectively.³ Furthermore, borohydride based $\text{Mg}(\text{BH}_4)_2\text{–NaBH}_4$ / DGM (diglyme) have shown to be chemically reactive by irreversibly modifying the pristine TiS₂ cathode² after the first discharge and lowered the anodic stability when the concentration of $\text{Mg}(\text{BH}_4)_2$ was increased. In addition, there is strong association between anion and cation in both Cl-based and borohydride-based electrolytes, generating a high energy barrier for dissociation at cathode-electrolyte interface. The respective monovalent cation species (such as MgCl^+) might even intercalate into the cathode as a whole, which makes it more complicated for exploring the cation co-intercalation strategy. In comparison, the dual salt electrolytes used in this work have superior anodic stability due to the strong C-F bond in $[\text{B}(\text{hfp}_4)]^-$ which render it as an ideal electrolyte system for exploring the cation co-intercalation strategy.⁴”

The desolvation of cations in the electrolyte solutions before intercalation can be different for Li, Na and Mg. Please, comment if the desolvation process influences electrochemistry.

Response: The de-/solvation process for cations with different charge densities can be different as has been reported previously (<https://doi.org/10.1039/C2CP40612H>). It was found that in carbonate-based solvents, Li^+ and Mg^{2+} are solvated strongly, whereas Na^+ is weakly solvated. In single ion systems, calculated desolvation energies were roughly the same for Li^+ and Na^+ (20 kcal/mol) at the surface of Li₄Ti₅O₁₂ (LTO) cathode. However, the desolvation of Mg^{2+} was significantly higher (roughly 40 kcal/mol) (<https://doi.org/10.1021/acsomega.1c04161>). Furthermore, Mg^{2+} exhibited stronger surface adsorption energy. In the case of dual-salt electrolytes, the scenario is different with the salts forming heteronuclear complexes or dual-cation clusters ($\text{M}_1^{\text{p}+} \text{M}_2^{\text{q}+} (\text{solvent molecule})_n$) due to the scarcity of solvent. The desolvation penalty of these heteronuclear dual-cation clusters is lower than the combined desolvation energy of the corresponding single cations with the same number of solvent molecules. Comparison of $\text{Mg}^{2+}\text{–Li}^+$ and $\text{Mg}^{2+}\text{–Na}^+$ dual-cation clusters in ethyl carbonate (EC) showed that the former had a higher desolvation energy than the later (<https://doi.org/10.1021/acsomega.1c04161>). Based on these findings we can comment that that desolvation process could affect the electrochemistry.

We have briefly stated this in the main text on page 27 as follows:

“Additionally, dual cation co-intercalation strategy executed by the dual salt electrolyte approach also has the possibility to lower the de-solvation penalty at the cathode-electrolyte interface, thus enhancing the electrochemical performance.⁵”

In Fig. 2b, the electrochemistry in Na-Mg-system should be further compared with the literature. For example, this reference: Bian et al. J. Mater. Chem. A, 2017,5, 600-608. <https://doi.org/10.1039/C6TA08505A>.

Response: We thank the reviewer for making us aware of this interesting publication. The electrochemical data of the Mg-Na system has now been compared with the data reported in the article suggested by the reviewer and necessary changes have been made in the manuscript on pages 7,8 as following:

“The electrochemical performance showed an initial discharge capacity of $\sim 250 \text{ mAh g}^{-1}$ (Fig. 2b), similar to the Mg-Li system. For comparison, the single salt Na system delivered $\sim 237 \text{ mAh g}^{-1}$ after the first discharge (**Fig. S5**). It is worth noting that the initial discharge capacity of the Mg-Na system also matched closely with the work by Bian and co-workers.² However, differences between the voltage profiles exist and could be traced to the chemical modification of TiS_2 cathode upon reacting with the borohydride-based dual salt electrolyte, while with the borate-based dual salt electrolyte, the TiS_2 did not exhibit chemical change. Following charging, the Mg-Na system provided a charge capacity of $\sim 196 \text{ mAh g}^{-1}$, which corresponds to a capacity loss of 54 mAh g^{-1} .”

In Fig. 2c, the electrochemical behavior of layered TiS_2 in Mg-system, is very different compared to the results shown in reference number 25 [Sun (2016)]. The electrochemistry of TiS_2/Mg reported by Linda Nazar’s group in reference 25 is better and the voltage-profile is different. This should be explained in the manuscript.

Response: The superior electrochemical performance of TiS_2 reported by Linda Nazar’s group was likely due to the battery being operated at a high temperature of $60 \text{ }^\circ\text{C}$. An explanation has been added in the manuscript on page 5 of the main text as follows:

“The poor performance of the single salt Mg cell was in contrast to the findings reported by the Sun *et al.*⁶ where an initial discharge capacity of $\sim 250 \text{ mAh g}^{-1}$ was delivered. The reason behind this could be that the measurements were conducted at an elevated temperature of $60 \text{ }^\circ\text{C}$.”

Lines 427-428 say that Li and Mg ions can potentially occupy either octahedral or tetrahedral sites in O1-TiS_2 . Accommodation of cations in tetrahedral sites of spinel TiS_2 has been reported by other authors. For example, see these references, and comment on them:

- *Chem. Mater.* 2018, 30, 7, 2436. <https://doi.org/10.1021/acs.chemmater.8b00552>.
- *Energy Environ. Sci.* 2016, 9, 2273. <https://doi.org/10.1039/C6EE00724D>

Response: We appreciate that the reviewer pointed us to these two insightful articles. The findings in both the suggested articles discussed the possibility of Mg occupying both tetrahedral and octahedral sites through mixed occupancy at non-dilute concentrations. Octahedral sites although are the preferred site of occupation at dilute concentrations. In case of Li storage, tetrahedral sites are not energetically favored for spinel TiS_2 . It is also worth mentioning, that mixed occupancy pertaining to Mg becomes energetically probable for concentrations in the range of $0.3 \leq x \leq 0.6$ for spinel TiS_2 . As the concentration approaches $x \sim 1$ the preferred occupancy reverts back to only octahedral site occupation in order to reduce cation-cation repulsion between the cations occupying the adjacent octahedral and tetrahedral sites. However, in our case the Mg concentration is still in the dilute regime, which suggests that octahedral site is the energetically preferred site.

We have cited the suggested articles and clarified the occupancy preference of Mg and Li in respect to layered TiS₂ on page 18 as follows:

“The site occupation is more nuanced in thiospinel Ti₂S₄ with Mg²⁺ exhibiting mixed occupancy at non-dilute Mg concentrations.^{7,8}”

In the XRD patterns (Fig. 3), after looking at the low-intensity reflections at around 20 ° two-theta, it seems to me that these two reflections are reversibly modified in the discharge/charge of the Mg-Li system (Fig. 3a), but this process is irreversible in the Mg-system (Fig. 3b). Please, comment on that.

Response: The reason the reflection at 20.2 ° appears to be reversible to a greater degree in the Mg-Li system compared to the Mg system is likely related to the extent of reversibility of the Mg ions. From our ICP-OES analysis, it can be seen that a slightly higher concentration of Mg ion was extracted from the TiS₂ in the Mg-Li system (Mg/Ti extracted = 0.24 – 0.17 = 0.07) than the Mg system (Mg/Ti extracted = 0.16 – 0.11 = 0.05). However, it should be noted that reflections at high angles are low in intensity and with poor resolution, so the interpretation can be inaccurate. In this case, the concerned reflection seems to reappear as a shoulder in the Mg system, which suggests some degree of reversibility. The same reflection in the Mg-Li system shows only a slightly better resolution, which agrees with the observation that substantial amount of Mg²⁺ got trapped after the 1st charge, similar to the Mg system.

*The theoretically calculated activation energy for Li and Mg migration is lower in Li_{0.5}TiS₂ (Fig. 8b) than TiS₂. I wonder if the energy reference (arbitrarily 0.0 eV in Fig. 8b) of Li_{0.5}TiS₂ is not higher than the energy reference of TiS₂, and this may be the reason for the apparent lower activation energy. Thus, Li_{0.5}TiS₂ could be like a sort of energized or entatic state as was recently proposed for certain electrodes [Pérez-Vicente. *PhysChemChemPhys* 25 (2023) 15600. DOI: 10.1039/d2cp06044b], due to cation-cation repulsions, with a smaller height of the hopping, compared to TiS₂. I would suggest calculating the energy of Mg site (octahedral) for TiS₂ and Li_{0.5}TiS₂.*

Response: We thank the reviewer for the insightful comment. The reference energies are relative to the energy of migrating cations (Li⁺, Na⁺, and Mg²⁺) present within the sixfold octahedral site (global minimum). Describing the reaction coordinate, we visualize a line that links the energy minima of the octahedral site and the tetrahedral site. The migration barrier, which is our focus, is evaluated by the energy difference between of Li⁺, Na⁺, and Mg²⁺ ions in an octahedral site and the transition-state site. The energy reference levels are not influenced by presence of Li ions, but caused by the rather slightly larger volume that guarantees the reliability of our outcomes.

Furthermore, the reviewer’s proposal to compute the energy associated with the Mg site (octahedral) in both TiS₂ and Li_{0.5}TiS₂ has captured our interest. Exploring the migration behavior of Mg in both materials possesses the potential to yield insights into their respective energetic characteristics and possible cation-cation interactions. We quantified the Mg intercalation energy (E_{int}^{Mg}) in both TiS₂ and Li_{0.5}TiS₂ with respect to the metallic magnesium anode. The aim was to

analyze the repulsion interactions between migrating cations. Our calculations revealed that these energies are comparable in both lithiated and delithiated compounds. Consequently, at the studied concentration, there is no perceivable cation-cation repulsion present.

Compound	E_{int}^{Mg} (eV)
TiS ₂	-2.82
Li _{0.5} TiS ₂	-2.86

Reviewer #2

This study aimed to overcome the limited redox activity of Mg ions in a TiS₂ cathode by introducing dual cation co-intercalation with faster Li ions, facilitated by a fluorinated alkoxyborate-based dual salt electrolyte. The concept was extended to an analogous system with Na ions instead of Li, yielding similar performance and achieving an initial discharge capacity close to the theoretical value. Through a combination of experimental and computational techniques, detailed explanations were provided for the divergent behavior observed between these two seemingly similar systems, including the charge storage mechanism and redox mechanism for the model system.

This research may have considerable implications for the advancement of high-voltage cathode materials. Therefore I would like to recommend this work, with the condition that the authors address the following comments in their revision:

Response: We thank the reviewer for recommending the work and are also appreciative of the constructive comments and suggestions. After carefully consideration we have tried to address all the concerns of the reviewer to the best of our abilities and have modified the manuscript accordingly.

In the abstract, the objective of the study was stated as the enhancement of Mg²⁺ mobility. However, this paper does not present any electrochemical performances, including rate performances.

Response: We appreciate the feedback from the reviewer, but some of the electrochemical performance data were not provided in the main script as the focus of the article was to critically discuss the scope and influence of dual cation co-intercalation on the insertion kinetics of Mg²⁺. We think that the galvanostatic charge discharge profiles of the Mg-Li, Mg-Na and Mg systems reflected the general improvement in the electrochemical performance. Furthermore, NEB calculations of the migration barriers indirectly showed an apparent enhancement in the mobility of the charge carriers in the dual cation systems.

However, it was the detailed elemental analysis using ICP-OES which enabled us to parse through the contribution of both cationic charge carriers and determine the true contribution of Mg²⁺ for the Mg-Li and Mg-Na systems. As discussed in the manuscript, the Mg-Li system showed improved Mg²⁺ insertion kinetics, whereas the Mg-Na system, despite much improved

electrochemical performance (similar to the Mg-Li system), exhibited severely suppressed Mg²⁺ intercalation and the system functioned as a *de-facto* sodium ion battery.

In this case we sincerely think that focusing on the absolute performance metrics won't allow us to accurately gauge and quantify the true impact on the reversible Mg²⁺ shuttle. Nevertheless, we have already provided cycling data of all the electrochemical systems discussed in this article as well as the discharge-charge voltage profile of the Mg, Li and Na reference systems in the Supporting Information (SI) in order to provide wider context to the reader. Moreover, as per the suggestion of the reviewer, we have now also added the rate capability data of the co-intercalating Mg-Li system in the SI (Fig. S2c).

In the dual-ion systems, the TiS₂ showed capacity fading in the initial cycles. In the extended cycles, the CE was around 90%. What are the possible reasons? In addition, why CE is greater than 100% in Figure S2?

Response: The capacity fading in the initial cycles is most likely due to the Mg entrapment during the initial cycles. This was also reported by (<https://doi.org/10.1021/acsenergylett.6b00145>) Sun *et al.* However, the reversibility of Mg ions was improved in the Mg-Li system.

The problem with the low Coulombic efficiency after extended cycling is likely due to side reactions between the electrolyte and the anode. However, the nature of interaction involving the dual salt electrolyte still needs further investigation.

As for the CE being greater than 100 % during the initial cycles (Figure S2), it is likely due to Mg entrapment. As the formula used to calculate Coulombic efficiency is $CE = \frac{\text{Discharge}}{\text{Charge}} * 100$, the initial cycles had CE > 100 % as a result of lower charge capacity than discharge capacity. This behavior has also been reported by Sun *et al.*

In the Mg-Na system, the Mg intercalation was impeded. Note, the TiS₂ was deeply discharged, with the possibility to trigger conversion reaction. Such severe structural change might also be detrimental to further intercalation. The authors should clarify this point.

Response: The reviewer is correct to point out that conversion reactions could be triggered at deep discharge states and that such reactions would inflict severe structural change which could strongly affect further intercalation. However, from *ex situ* XRD we did not see any potential conversion products at least in the bulk, as previously reported (<https://doi.org/10.1002/advs.201801021>). Although, it is possible for such reactions to occur on the surface. It should also be noted that the voltage profile and capacity obtained on full discharge (ca. 250 mAh g⁻¹, 1e⁻) corresponded to the intercalation regime. The following work (<https://doi.org/10.1021/acsnano.9b04222>) done by Wang *et al.* showed that conversion reaction resulted in > 1e⁻ exchange. The part of the voltage profile which relates to the pure intercalation as per this report, agreed very well with our voltage profile upon deep discharge. Hence, we infer that although conversion is a possibility, the reaction in this case is intercalation dominant. Additionally, elemental analysis even at 1.3 V and 0.6 V in the first discharge showed Na to be dominant charge carrier, suggesting that potential conversion reaction was not the obvious reason for suppressed Mg²⁺ intercalation.

However, the TiS_2 structure is likely to get distorted when trying to accommodate Mg^{2+} after undergoing Na^+ induced phase change (P3 and O3), which created an unfavorable coordination geometry. This is probably the critical reason for the impeded intercalation of Mg^{2+} .

What are the difference or advantages of the cation co-intercalation concept over other co-intercalation approaches reported before? It's recommended to discuss it as perspective or outlook.

Response: The co-intercalation approaches which have been employed specifically in Mg batteries are primarily solvent-cation co-intercalation. The concept was reasonably successful as this strategy could lower the polarization of Mg^{2+} by shielding the charge and thus ensuring faster transport. In the category of solvent-cation co-intercalation, charge shielding via hydration shell has been used in certain cases. Although, the cation transportation improved, the main drawback of this approach is the passivation of the Mg anode (<https://doi.org/10.1039/C4CP05591H>). Another example of co-intercalation was reported by Yoo *et al.* (<https://doi.org/10.1038/s41467-017-00431-9>) where $[\text{MgCl}]^+$ was rapidly de-/intercalated in an expanded TiS_2 . A common issue of the above mentioned, either solvent-cation co-intercalation or cation-anion co-intercalation strategy, is the requirement of large quantity of electrolyte to store the charge carrier $[\text{Mg}(\text{solvent})_x]^{2+}$ or $[\text{MgCl}]^+$, thus lowering the energy density. Furthermore, the modified charged carriers are much larger in size compared to bare Mg^{2+} , normally requiring additional crystal engineering strategies to enlarge the storage sites.

In the dual cation approach, we are aiming at building a 'rocking chair' model whereby both cations are accommodated in the cathode during discharge and co-deposited on the anode during charge. This strategy does not require large quantity of electrolyte, thus not hampering the overall energy density while enabling a practical cell design.

As per the recommendation of the reviewer we have discussed the differences/advantages in greater detail in the 'Discussion' section, the amendments made on pages **26, 27** are as follows:

“Typically, the dual salt electrolyte approach has been tuned towards designing hybrid battery systems, where the cathode only accommodates the faster Li^+/Na^+ (discharge) while the slower Mg^{2+} plates on the metal anode (charge). Effectively, the cathode operates as a Li^+/Na^+ pass filter.¹ This strategy has been reported to be a viable way to circumvent the problematic kinetics of Mg^{2+} . However, as the charge carriers (all Mg^{2+} after discharge and all Li^+ after charge) are stored in the electrolyte solution, large amount of solvent is required, which consequently lowers the energy density. Not to mention, the cell design will need to be altered to safely hold larger quantities of electrolyte. On the contrary, dual cation co-intercalation enabled by dual salt electrolyte is based on the 'rocking chair' principle.⁹ The advantage of this approach is that the charge carriers are accommodated in the electrodes and not in the electrolyte, thus ensuring higher overall energy density.

Co-intercalation approaches have been investigated as an alternative counter measure to combat the sluggishness of Mg^{2+} . Water solvated Mg^{2+} was first studied by Song *et al.* to shield

the charge and decrease the polarization.¹⁰ Although, the capacity was much improved, the presence of water in the electrolyte passivated the Mg anode.¹¹ Along similar lines, V₂O₅ xerogels with interlayer water molecules screened the intercalated Mg²⁺ and reduced the polarization. However, it was later reported that water molecules get extracted on charging along with Mg²⁺ causing the V₂O₅ structure to collapse. A more effective solvent co-intercalation method was reported by Li *et al.* with DME solvated [Mg • 3DME]²⁺ exhibiting fast kinetics in layered MoS₂ by shielding the high charge density of Mg²⁺.¹² A study done by Yoo *et al.* demonstrated fast kinetics in TiS₂ by the intercalation of monovalent [MgCl]⁺, which can be interpreted as the co-intercalation of Mg²⁺ and Cl⁻ ions.¹³ Nevertheless, the intercalation of such a bulky specie, either [Mg(solvent)_x]²⁺ or [MgCl]⁺, could cause steric hindrance and hence require artificial modification of the host structure. Furthermore, cation-solvent co-intercalation or cation-anion co-intercalation requires large amount of electrolyte as reservoir of solvents or anions, which in turn will affect the energy density.

The co-intercalation concept reported in this work corresponds to the simultaneous accommodation of two cations in the host crystal structure without the need for any artificial structural modification of the cathode due to the much smaller size of cationic charge carriers. Thus, ensuring greater structural integrity and stability. Additionally, dual cation co-intercalation strategy executed by the dual salt electrolyte approach also has possibility for lower de-solvation penalty at the cathode-electrolyte interface, thus enhancing the electrochemical performance.⁵ Furthermore, compared to the other co-intercalation methods, electrolyte amount is not a rate limiting parameter for achieving higher energy densities. Finally, both cations are active charge carriers and enable obtaining high specific capacities.”

The authors claimed that "activation energies (E_a) were 0.60 and 1.28 eV for Li and Mg ions, respectively, as depicted in Fig. 8b (left)". However, this claim is inconsistent with the diffusion barrier shown in Fig. 8b, where the diffusion barrier for Li is approximately 0.70 eV. This discrepancy requires clarification. Furthermore, the authors estimated the diffusion coefficient of Mg ions to be 10⁷ orders lower than that of Li based on DFT results. The method used for this estimation should also be clarified.

Response: We thank the reviewer for bringing this inconsistency to our attention. The discrepancy highlighted stems from a typographical error in our manuscript. The correct activation energy for Li ions is indeed approximately 0.73 eV, as indicated by the diffusion barrier shown in Fig. 8b. We have made the necessary corrections in our manuscript to ensure that this inconsistency is addressed (pages 19, 20).

Additionally, we understand the concern about the estimation of the diffusion coefficient for Mg ions being reported as 10⁷ orders of magnitude lower than that of Li. This estimation was based on our DFT (Density Functional Theory) calculations. Our computed activation energies (E_a) are 0.73, 1.07 and 1.23 eV for Li, Na and Mg ions, respectively, which translate to the 10⁻¹³, 10⁻¹⁸, and 10⁻²¹ diffusion coefficient (D) of Li, Na, and Mg ions respectively at room temperature, according to the standard Arrhenius expression $D \sim \exp(-E_a/k_B T)$. At room temperature, k_BT is 0.026 eV. We

have provided the details about the aforementioned methodology used for this estimation in our manuscript too. The revised version of the text, providing an explanation of the approach, has been added on the page 19 as follows:

“These values can be used to compute the diffusion coefficient (D) for the ions at room temperature ($k_B T = 0.026$ eV) using the established Arrhenius expression $D \sim \exp(-E_a/k_B T)$. Specifically, the calculated diffusion coefficients of Li and Mg ions at room are approximately 10^{-13} and 10^{-21} cm^2/s , respectively. This translates to a D of Mg ions that is 10^8 times lower than that of Li ions which underlined the significant sluggishness of Mg ions.”

The authors compared the charge difference of Mg, Li, and Na in Figure 8. To ensure a fair comparison at the same level, the isovalues for each of them need to be added. Additionally, the authors could strengthen their claim by conducting a charge analysis, which would provide a quantitative comparison of the ions.

Response: We completely agree that ensuring a fair comparison is crucial in our analysis of the charge difference among Mg, Li, and Na. To address this concern, we took the reviewer’s suggestion into account and added isovalues ($0.004 \text{ e}\text{\AA}^{-3}$) in regards to Figure 8 (page 18, 19). This provides a clearer visual representation of the charge difference at the same level, allowing for a more accurate comparison.

Furthermore, we understand the importance of a quantitative analysis to strengthen our claims. We have conducted a comprehensive charge analysis to quantitatively compare the ions, as has been suggested. The explanation is included in the supporting information as per the last remark of the reviewer.

The calculated energy barrier for Li intercalation in TiS_2 was ~ 600 meV, which is close to the threshold for intercalation chemistry (625 meV). Experimentally, the kinetics of Li^+ in TiS_2 was fairly good. What made the discrepancy?

Response: We thank the reviewer for the insightful question regarding the calculated energy barrier for Li intercalation in TiS_2 and the observed experimental kinetics of Li^+ in the same material. The apparent discrepancy between the calculated energy barrier and the experimental kinetics can be attributed to the temperature and environment. The experimental conditions, such as temperature and the surrounding environment, can play a significant role in intercalation kinetics. These factors might not have been fully accounted for in the theoretical calculations, which were done at 0 K. Higher temperature can cause volume expansion. The activation barriers governing ion diffusion within layered transition metal dichalcogenides (TMDs) exhibit sensitivity to changes in interlayer spacing (As mentioned before <https://doi.org/10.1021/acs.inorgchem.5b00188>, and <https://doi.org/10.1103/PhysRevB.78.104306>).

Furthermore, the experimental system could have defects, imperfections, or impurities that lower the effective energy barrier. These factors can create localized regions where intercalation is more favorable, leading to a faster observed kinetics. The intercalation process is inherently electrochemical in nature. Factors like charge transfer kinetics, double-layer capacitance, and mass

transport can influence the experimental kinetics and may not have been fully considered in the theoretical model.

As the pristine TiS₂ was with a O1 phase, the activation energy for Na⁺ migration in O1-TiS₂ should be added.

Response: In response to the reviewer's comment, we agree that including the activation energy for Na⁺ migration in the O1-TiS₂ phase is important for better comparative understanding of the transportation behavior of different charge carriers in the bulk of the material. We have incorporated the minimum energy path, as shown below in the supporting information (Fig. S10) and mentioned it in page 19. Based on the calculations, the E_a is ca. 1.07 eV in O1-TiS₂, hence between the migration barrier of Mg²⁺ on the higher side and Li⁺ on the lower side. However, we would like to emphasize that due to the bigger size of Na⁺, the O1-TiS₂ structure is not maintained in practice and that is why we focused our discussion, in the manuscript, on the migration barriers in the P3 and O3 structures.

Figure 1: The calculated activation barriers of Mg, Na, and Li ion diffusion in bulk a) O1-TiS₂ and b) O3-Li_{0.5}TiS₂.

The analysis of the phase diagram presented in the text is not easily comprehensible, particularly for the sodium system. Therefore, it is advisable to rephrase the paragraph in a more comprehensive manner, ensuring clarity and understanding.

Response: As per the reviewer's request we have rephrased the section discussing the phase diagram of the sodium system and tried to make it easier to understand. Changes made to the manuscript are reflected in page 22 of the manuscript are as follows:

“However, only two-phase stable ternary compounds exist, in the form of NaTiS_2 and $\text{Na}_{0.3}\text{TiS}_2$ (dark green points) along with the metastable $\text{Na}_{0.5}\text{TiS}_2$ phase (red point). The metastable nature of $\text{P3-Na}_{0.5}\text{TiS}_2$ thermodynamically drives the conversion towards the phase stable O3-TiS_2 . In order to clarify the role that the stability of the structure played to inhibit the intercalation of Mg ions in the Mg-Na system, the structural stability of the P3 and O3 phase was calculated after the incorporation of Mg ions. The outcome was that significant distortions were induced in the TiS_2 crystal structure (Fig. 10d). It was seen that the accommodation of Mg ions in O3-TiS_2 generated relatively stronger distortions compared to the metastable P3-TiS_2 (Table S3). Thus, the transition of the metastable P3 phase to the thermodynamically stable O3 phase is likely to exacerbate the overall distortions in the crystal lattice, leading to severe structural destabilization. This phenomenon explained well the suppression of Mg^{2+} co-intercalation, as well as the structural degradation observed with cycling (Fig. S6). On the flipside, O1-TiS_2 phase is retained when storing the smaller Li ions, ensuring no phase change. Therefore, no distortions were introduced while accommodating Mg ions in the O1 phase (Fig. 10d) due to similarity of ionic radius.”

When examining the redox activity of TiS_2 , relying solely on the calculated DOS may not be sufficient to unveil the bonding/anti-bonding states. Additional methods such as COHP analysis should be incorporated to provide a more comprehensive understanding. Furthermore, it is suggested that charge transfer analysis be included to support the assertion that TiS_2 does not exhibit pronounced sulfur redox activity but rather demonstrates Ti as the primary redox active site when charge carriers are intercalated.

Response: We thank the reviewer for the insights and agree that a comprehensive examination of the redox activity of TiS_2 is crucial for a thorough understanding of its properties. We have incorporated an additional method in COHP (Crystal Orbital Hamilton Population) analysis as per the reviewer’s request, to further elucidate the bonding/anti-bonding states within the material, as shown below. Furthermore, we applied charge transfer analysis in supporting our assertion about the primary redox active site in TiS_2 when charge carriers are intercalated. We included this analysis to provide a more robust and well-rounded picture of the redox behavior of TiS_2 . It should also be noted that from the EELS spectra, no discernible S redox was observed which also strongly indicated to a Ti dominant redox.

Figure 2: The Crystal Orbital Hamiltonian (COHP) analysis between a Ti-d orbital and S-p state. COHP values manifest as both negative (indicating bonding, shown in red) and positive (indicating anti-bonding, shown in blue) interactions, positioned below and above the horizontal black line, respectively. The zero energy is aligned with the valence band top.

Table 1: The calculated charges using PBE functional.

Element	Charge (vasp) TiS ₂	Charge (bader) TiS ₂	Charge (vasp) LiTiS ₂	Charge (bader) LiTiS ₂
Li	-	-	2.074	2.129
Ti	8.574	8.291	8.585	8.427
S	3.745	6.854	3.795	7.221
S	3.745	6.855	3.795	7.221

Table 2: The calculated charges using HSE06 hybrid functional.

Element	Charge (vasp) TiS ₂	Charge (bader) TiS ₂	Charge (vasp) LiTiS ₂	Charge (bader) LiTiS ₂
Li	-	-	2.067	2.116
Ti	8.378	7.997	8.506	8.230
S	3.804	7.002	3.844	7.326
S	3.804	7.001	3.844	7.327

Experimentally, cycling within an anodic voltage range sufficiently high, the process of lithiation/delithiation in TiS₂ is highly reversible, and structural alterations are kept to a minimum (<https://doi.org/10.1021/jacs.2c02668>). These changes are primarily seen as a slight lengthening of the Ti–S bond, causing minimal structural adjustments. This places intercalation reactions on the far left of the anti-bonding continuum (blue) shown in COHP plot, where redox contributions from the anion are relatively small, and correspondingly, structural modifications are minimal.

Additionally, the sharp peak in the bonding continuum (red), consisting primarily of the S-p orbitals, vanish, due to the weakening of the Ti–S bonding upon the intercalation of Li. The broadening of the anti-bonding continuum observed near the fermi level corresponds to delocalization of electrons in the empty Ti-3d orbitals.

Regarding the charge analysis, Zhang *et al.* (<https://doi.org/10.1021/acs.nanolett.8b01680>) performed Bader charge analysis, which calculates the charge on each atom, and predicted that one Ti and two S gain 0.089 and 0.384 electrons, respectively, for each intercalated Li⁺, indicating that S underwent redox mechanism. However, it's important to note that no single DFT functional is universally superior for all types of systems or properties. The choice of functional should be based on the specific goals of the research and a careful assessment of its performance for the particular system of interest. In our Bader charge analysis, we found that one Ti and two S gain 0.136 and 0.366 electrons, respectively using the PBE functional, while the more accurate functional such as HSE06¹⁴ shows one Ti and two S gaining 0.233 and 0.324 electrons. However, VASP code also listed 0.011 and 0.050 gaining electrons for Ti and S respectively using PBE functional while for HSE06 functional predicted 0.128 and 0.040 gaining electrons for Ti and S. In fact, one can of course always argue about the reliability of charge analysis, but within the computational setup used, the Ti redox becomes more prominent when using the more accurate functional. Although, the process of charge compensation in TiS₂ exhibits a degree of shared mechanism between Ti and S, based on charge analysis, it is important to note that the layered structure of TiS₂ is thermodynamically stable, which means that the removal of lithium ions (delithiation) from LiTiS₂ has minimal impact on the S sublattice, despite the fact that S in TiS₂ is undercoordinated. Therefore, the contributions from S atoms are relatively minor.

In the response to the remark of the reviewer, discussion of COHP and charge analysis have been included and can be found in pages 24, 25 while the associated figure and the tables have been added to the supporting information (Fig. S11 and Tables S4 and S5). The following part has been added to the main text:

“The bonding/anti-bonding states were further analyzed using the Crystal Orbital Hamiltonian (COHP) analysis as can be seen in **Fig. S11**. It can be observed that upon reduction, the continuum corresponding to the anti-bonding states (blue) near fermi level underwent broadening due to the delocalization of the electrons in the empty Ti-d states. Simultaneously, in the bonding continuum (red), which is primarily comprised of the S-p orbitals, the sharp peak vanishes as the bonding weakens due to slight lengthening of the Ti–S bond on Li⁺ intercalation. These observations complement well with the DOS calculations and suggest that Ti is the dominant redox site.

Moreover, charge analysis was also conducted to support the assertion that TiS₂ predominantly undergoes redox at the Ti site. It should be noted that in the study done by Zhang *et al.* and the subsequent Bader charge analysis, it was found that one titanium (Ti) atom and two sulfur (S) atoms gain electrons when lithium ions are intercalated in TiS₂.¹⁵ They predicted that both Ti and S contribute to the charge compensation mechanism, with S gaining more electrons than Ti upon intercalation of one Li⁺ per formula unit. In our study, we found that PBE functional predicts incorrectly the charge compensation process in TiS₂ which involves a shared mechanism between

Ti and S with more contribution from S (Table. S4). However, the Ti cations gain more electrons using the more accurate HSE06 hybrid functional¹⁴ and the contribution of the S anions to the charge compensation process becomes relatively small (0.128 for Ti and 0.040 for S), following the charges listed in the VASP code (Table. S5).”

References

- 1 Gao, T. *et al.* Hybrid Mg²⁺/Li⁺ Battery with Long Cycle Life and High Rate Capability. *Advanced Energy Materials* **5**, 1401507, doi:<https://doi.org/10.1002/aenm.201401507> (2015).
- 2 Bian, X. *et al.* A long cycle-life and high safety Na⁺/Mg²⁺ hybrid-ion battery built by using a TiS₂ derived titanium sulfide cathode. *Journal of Materials Chemistry A* **5**, 600-608, doi:10.1039/C6TA08505A (2017).
- 3 Yagi, S. *et al.* A concept of dual-salt polyvalent-metal storage battery. *J. Mater. Chem. A* **2**, 1144-1149, doi:10.1039/C3TA13668J (2014).
- 4 Zhao-Karger, Z., Gil Bardaji, M. E., Fuhr, O. & Fichtner, M. A new class of non-corrosive, highly efficient electrolytes for rechargeable magnesium batteries. *Journal of Materials Chemistry A* **5**, 10815-10820, doi:10.1039/C7TA02237A (2017).
- 5 Rasheev, H., Stoyanova, R. & Tadjer, A. Rivalry at the Interface: Ion Desolvation and Electrolyte Degradation in Model Ethylene Carbonate Complexes of Li⁺, Na⁺, and Mg²⁺ with PF₆⁻ on the Li₄Ti₅O₁₂ (111) Surface. *ACS Omega* **6**, 29735-29745, doi:10.1021/acsomega.1c04161 (2021).
- 6 Sun, X., Bonnicksen, P. & Nazar, L. F. Layered TiS₂ Positive Electrode for Mg Batteries. *ACS Energy Letters* **1**, 297-301, doi:10.1021/acseenergylett.6b00145 (2016).
- 7 Kolli, S. K. & Van der Ven, A. First-Principles Study of Spinel MgTiS₂ as a Cathode Material. *Chemistry of Materials* **30**, 2436-2442, doi:10.1021/acs.chemmater.8b00552 (2018).
- 8 Sun, X. *et al.* A high capacity thiospinel cathode for Mg batteries. *Energy & Environmental Science* **9**, 2273-2277, doi:10.1039/C6EE00724D (2016).
- 9 Li, H. *et al.* Fast Diffusion of Multivalent Ions Facilitated by Concerted Interactions in Dual-Ion Battery Systems. *Advanced Energy Materials* **8**, 1801475, doi:<https://doi.org/10.1002/aenm.201801475> (2018).
- 10 Song, J. *et al.* Activation of a MnO₂ cathode by water-stimulated Mg²⁺ insertion for a magnesium ion battery. *Physical Chemistry Chemical Physics* **17**, 5256-5264, doi:10.1039/C4CP05591H (2015).
- 11 Novák, P. & Desilvestro, J. Electrochemical Insertion of Magnesium in Metal Oxides and Sulfides from Aprotic Electrolytes. *Journal of The Electrochemical Society* **140**, 140, doi:10.1149/1.2056075 (1993).
- 12 Li, Z. *et al.* Fast kinetics of multivalent intercalation chemistry enabled by solvated magnesium-ions into self-established metallic layered materials. *Nature Communications* **9**, 5115, doi:10.1038/s41467-018-07484-4 (2018).
- 13 Yoo, H. D. *et al.* Fast kinetics of magnesium monochloride cations in interlayer-expanded titanium disulfide for magnesium rechargeable batteries. *Nature Communications* **8**, 339, doi:10.1038/s41467-017-00431-9 (2017).
- 14 Krukau, A. V., Vydrov, O. A., Izmaylov, A. F. & Scuseria, G. E. Influence of the exchange screening parameter on the performance of screened hybrid functionals. *The Journal of Chemical Physics* **125**, doi:10.1063/1.2404663 (2006).
- 15 Zhang, L. *et al.* Tracking the Chemical and Structural Evolution of the TiS₂ Electrode in the Lithium-Ion Cell Using Operando X-ray Absorption Spectroscopy. *Nano Letters* **18**, 4506-4515, doi:10.1021/acs.nanolett.8b01680 (2018).

REVIEWER COMMENTS

Reviewer #1 (Remarks to the Author):

In general, the modifications of the revised manuscript are satisfactory. Actually, I have new doubts about the claimed intercalation of Mg in the Li-Mg-TiS₂ system, simultaneously with Li-intercalation. It could be that only Li is intercalated, and Mg could be rather accommodated in the particle surface. The authors discard a conversion reaction in the Mg-TiS₂ system, but accommodation of Mg in the particle surface should be also considered.

Irrespective of the theoretical calculations, the claimed co-intercalation of Mg-Li is mainly based on the elemental analysis of the electrodes, but the electrodes could be contaminated by traces of the electrolyte. In the Mg-TiS₂ system, no change of the XRD patterns is observed upon Mg-intercalation, in contrast to Li-TiS₂ and Li-Mg-TiS₂ systems.

The authors say that phase change was not observed in the Raman spectra for Mg intercalation in the Mg-TiS₂ system (Fig. 5b). If one looks at the Raman spectra (Fig. 5 and Fig. S7), it seems that there is not Mg intercalation into TiS₂, even in the Mg-Li-TiS₂ system, just Li intercalation.

The reasons for claiming simultaneous co-intercalation of Mg and Li, based on experimental results, should be more clearly stated in the Discussion section.

Reviewer #2 (Remarks to the Author):

The authors have appropriately addressed my concerns, and I am delighted to endorse the publication in Nature Communications.

Reviewer #1

In general, the modifications of the revised manuscript are satisfactory. Actually, I have new doubts about the claimed intercalation of Mg in the Li-Mg-TiS₂ system, simultaneously with Li-intercalation. It could be that only Li is intercalated, and Mg could be rather accommodated in the particle surface. The authors discard a conversion reaction in the Mg-TiS₂ system, but accommodation of Mg in the particle surface should be also considered.

Irrespective of the theoretical calculations, the claimed co-intercalation of Mg-Li is mainly based on the elemental analysis of the electrodes, but the electrodes could be contaminated by traces of the electrolyte. In the Mg-TiS₂ system, no change of the XRD patterns is observed upon Mg-intercalation, in contrast to Li-TiS₂ and Li-Mg-TiS₂ systems.

The authors say that phase change was not observed in the Raman spectra for Mg intercalation in the Mg-TiS₂ system (Fig. 5b). If one looks at the Raman spectra (Fig. 5 and Fig. S7), it seems that there is not Mg intercalation into TiS₂, even in the Mg-Li-TiS₂ system, just Li intercalation.

The reasons for claiming simultaneous co-intercalation of Mg and Li, based on experimental results, should be more clearly stated in the Discussion section.

Response: We are glad to learn that explanations provided to the questions posed in the first round of the revision satisfied the reviewer and that they found the modifications adequate. In addition, we also treat the concern of the reviewer regarding the co-intercalation of Mg and Li into the structure of the TiS₂ with utmost respect and have tried to address these doubts by providing logical arguments to the best of our abilities.

The reviewer highlights that Mg is quite likely to be accommodated on the surface of the TiS₂ particles and we agree that this is a possibility, especially in the simple Mg-TiS₂ system due to the sluggish kinetics of Mg²⁺. However, in the presence of Li⁺ we found out that the kinetics of Mg²⁺ was superior and this in turn is likely to improve the bulk redox activity of Mg²⁺ in the Mg-Li-TiS₂ system specifically. The improved redox observed in TiS₂ in the Mg-Li system was also observed from the CV profile in Fig. 2d. Moreover, this CV profile is distinctly different to the profile for only Li de-/intercalation (Fig. S3b) suggesting that the redox behavior observed in the Mg-Li-TiS₂ system does not bear an obvious signature of only Li⁺ intercalation. The CV of the pure Mg-TiS₂ system on the other hand exhibited no obvious redox peaks which agrees well with the hypothesis that due to the slower kinetics of Mg²⁺ in the Mg-TiS₂ system, Mg²⁺ is likely getting accommodated on the surface and not penetrating into the structure sufficiently. The comparison of the CVs indirectly suggests to a redox involving both Mg²⁺ and Li⁺.

Elemental analysis showed that the concentration of Mg improved substantially in the Mg-Li-TiS₂ system (0.24 mol fraction of Mg in the Mg-Li system while 0.16 mol fraction in the pure Mg system per formula unit). If we presume that almost all of the Mg²⁺ carriers are reacting on the surface, that would also mean a much higher local areal Mg concentration. This is thermodynamically unstable for TiS₂ as can be seen in our calculated phase diagram for 1Mg per TiS₂ (Fig. 10b).

Addressing the concern of the reviewer about the concentration of Mg coming from contamination from the electrolyte, we actually observed a change in Mg concentration when charged after

discharge. We investigated the concentration change on cycling for the 1st and the 5th cycles, by carrying out ICP measurement on the cycled cathodes after the same treatment (washing by DME). This indicates that the Mg quantified in the samples were not trace remnants from the electrolyte.

The XRD comparison of TiS₂ between the Mg-TiS₂ and Mg-Li-TiS₂ systems was intended to highlight that both systems exhibited no phase change but a volume expansion was observed in the latter. The purpose of this characterization was simply to analyze the bulk structure and its evolution with de-/intercalation. The reason behind the discernible interlayer expansion in the Mg-TiS₂ system is understandable as on its own Mg²⁺ is slow and the concentration of Mg per TiS₂ was also dilute. However, in the co-intercalating Mg-Li-TiS₂ system, the concentration of Mg and Li after 1st discharge was 0.24 and 0.35 mol fractions i.e. a combined concentration of 0.59 per TiS₂ formula unit. This is a substantial increment compared to the 0.16 mol fraction which was intercalated in the Mg-TiS₂ system. The far greater concentration of charge carriers in the Mg-Li-TiS₂ system is the reason for observed expansion of the interlayer distance.

It is worth mentioning, that Sun *et al.* <https://doi.org/10.1021/acsenergylett.6b00145> reported that phase transition of TiS₂ was triggered when the Mg concentration reached ca. 0.2 mol per formula unit with the cell being operated at 60 °C. In our work we do not observe any phase transition in the Mg-Li system with the Mg concentration at 0.24 and this is likely due to the presence of Li which stops such a phase change and maintains the solid-solution type behavior of TiS₂ as can also be observed in the smooth voltage profile of Fig. 2a.

In regard to the local structure investigation of the Mg-Li-TiS₂ with Raman spectroscopy, it is not possible to confirm whether Mg, Li or both charge carriers are intercalating just from the fitted spectra themselves. The general observation was that Mg-Li-TiS₂ underwent local structure modification as can be seen from the broadening of the out-of-plane A_{1g} and Sh peaks in Fig. 5b. The corresponding Mg-TiS₂ system showed no obvious change which is in agreement with the XRD. We appreciated that just the spectra themselves were not strong enough evidence to confirm the co-intercalation of Mg²⁺ and Li⁺ into TiS₂. Hence, detailed analysis was done by plotting a distribution of the vibrational frequencies for different test or reference systems to check if any trend developed.

In Fig. 6b, c we plotted the A_{1g} and Sh modes for Mg-Li-TiS₂ and compared the vibrational modes with pristine TiS₂, Mg-TiS₂ (full discharge), Li-TiS₂-mid discharge (half-discharge) and LiTiS₂ full discharge. All discharge samples are colour coded in yellow with red borders. When we look at the distribution, we see that there is an increase in vibrational frequency (blue shift) going from Mg-TiS₂ to Mg-Li-TiS₂. The increment in out-of-plane vibrational frequency corresponds to increasing resistive force upon intercalation. It is important to reiterate that the Mg and Li concentration in Mg-Li-TiS₂ were 0.24 and 0.35 respectively, whereas for a fully and half discharged Li-TiS₂ system, the concentration of Li should in principle be 0.5 and 1 mol per TiS₂ unit, respectively. In our investigation we observed that the A_{1g} and Sh modes of the Mg-Li system exhibited a substantially higher vibrational frequency (stronger resistive force) compared to the half-filled Li-TiS₂ system (Fig. 6b, c). In fact, when compared to the fully lithiated Li-TiS₂, the Mg-Li-TiS₂ system showed a stronger blue shift by ca. 8 cm⁻¹. If we assumed only Li intercalation in the Mg-Li-TiS₂ system, that would mean that 0.35 mol fraction of Li in TiS₂ inflicted a much higher resistive force compared to the half-lithiated (0.5 mol Li) Li-TiS₂ system (half-discharge),

which is paradoxical. However, in conjunction with 0.24 mol of Mg, the concentration of charge carriers (Mg^{2+} and Li^+) is ca. 0.6 per TiS_2 with a total charge of 0.83 (0.48 from Mg^{2+} and 0.35 from Li^+) and this explains that Li^+ together with Mg^{2+} increases the vibrational frequency. The only way this can happen is if both Li^+ and Mg^{2+} intercalate into the structure of TiS_2 .

Some of the arguments and discussions were already made. As per the reviewer's suggestion, we have further clarified the co-intercalation in the Discussion section in page 28. The modification made in the manuscript is as follows:

“Raman spectroscopy showed the interaction of the intercalated ions with the surrounding sulfur atoms. Comparing the vibrational modes of the Mg-Li system with the half-discharged and fully discharge Li systems further highlighted that Mg and Li ions were co-intercalated into the structure. This was clarified as a stronger interaction was observed with the surrounding sulfur atoms in the Mg-Li system, comprising of 0.35 Li together with 0.24 Mg, in comparison to the half lithiated Li system (half-discharged). Moreover, when compared against the fully lithiated Li system, the Mg-Li system still exhibited a slightly stronger blue shift in the out-of-plane frequencies by 8 cm^{-1} likely due to stronger Coulombic interaction due to the presence of divalent Mg^{2+} . The above specified observations lends credence towards improved storage of Mg^{2+} in tandem with Li^+ in the Mg-Li system, whereas the single ion Mg system suffered to sufficiently intercalate Mg^{2+} and showed no obvious change in the Raman modes due to dilute Mg concentration in the TiS_2 .”

REVIEWERS' COMMENTS

Reviewer #1 (Remarks to the Author):

After reading the revised manuscript and the authors' response, I think that the manuscript can be recommended for publication in Nature Communications in its actual state.